**Mass concentration measurements of autumn bioaerosol using low cost sensors in a mature temperate woodland Free Air Carbon Dioxide Enrichment (FACE) experiment: investigating the role of meteorology and carbon dioxide levels**

Aileen B. Baird[1,2], Edward J. Bannister[1,2], A. Robert MacKenzie[1,2], and Francis D. Pope[1,2*]

[1]School of Geography, Earth & Environmental Sciences, University of Birmingham, Birmingham, B152TT, UK
[2]Birmingham Institute of Forest Research, University of Birmingham, Birmingham, B152TT, UK

*Correspondence to*: Francis Pope f.pope@bham.ac.uk

**Abstract.**

Forest environments contain a wide variety of airborne biological particles (bioaerosols), including pollen, fungal spores, bacteria, viruses, plant detritus and soil particles. Forest bioaerosol plays a number of important roles related to plant and livestock health, human disease and allergenicity, forest and wider ecology, and are thought to influence biosphere/atmosphere interactions via warm and cold cloud formation. Despite the importance of bioaerosols, there are few measurements of forest aerosol, and there is a lack of understanding of how climate change will affect forest bioaerosol in the future.

We installed low-cost optical particle counters (OPCs) to measure particles in the size range between 1 and 10 μm ($PM_{10}$-$PM_1$) for a period of two months in Autumn 2018 at the Birmingham Institute of Forest Research (BIFoR) Free Air Carbon Dioxide Enrichment (FACE) facility. In the paper, we propose that the $PM_{10}$-$PM_1$ metric is a good proxy for bioaerosols because of the bioaerosol representative size range, the location of the study site (a woodland in a rural location), the field measurement taking place during the season of peak fungal activity, and the low hygroscopicity of the particles measured. The BIFoR FACE

facility fumigates three 700 $m^2$ areas of the forest with an additional 150 ppm $CO_2$ above ambient with minimal impacts on other potential environmental drivers such as temperature, humidity, and wind. This experimental set-up enabled us to investigate the effect of environmental variables, including elevated $CO_2$ ($eCO_2$), on bioaerosol proxy concentrations, and to evaluate the performance of the low-cost OPCs in a forested environment.

Operating the low-cost OPCs during Autumn 2018, we aimed to capture predominantly the fungal bioaerosol season. Across

the experimental duration, the OPCs captured both temporal and spatial variation in bioaerosol concentrations. Aerosol concentrations were affected by changing temperatures and wind speeds, but, contrary to our initial hypothesis, not by relative humidity. We detected no effect of the $eCO_2$ treatment on total bioaerosol concentrations, but a potential suppression of high concentration bioaerosol events was detected under $eCO_2$. In-canopy atmospheric dispersion modelling indicates that the median spore dispersion distance is sufficiently small that there is little mixing between treatment and control experiments.

Our data demonstrate the suitability of low-cost OPCs, interpreted with due caution, for use in forests, and so opens the possibility of forest bioaerosol monitoring in a wider range of habitats, to a wider range of researchers at a modest cost.

## 1 Introduction

Aerobiology investigates the production, transport, and removal of airborne biological particles such as bacterial, fungal, and viral microbes, plant pollen, and soil and plant debris. Fungal spores form a large proportion of the bioaerosol population, with ground level concentrations typically 10,000–50,000 spores $m^{-3}$ (compared with 1000–2000 pollen grains $m^{-3}$) (Sesartic and Dallafior, 2011, and references therein). Modelling suggests that fungal spores form approximately 23% of bioaerosol mass globally (Heald and Spracklen (2009). Understanding the dispersal of fungal spores is not only essential for understanding fungal ecology and their direct relevance to the forest plant and soil communities, but also for understanding human disease and allergens, plant and crop health, livestock health, as well as the role played by bioaerosols in ice nucleation and cloud condensation (Pöschl, 2005; Reinmuth-Selzle et al., 2017; Sesartic and Dallafior, 2011a). The significant influence of airborne fungal particles on forest biology is often overlooked.

Spores are produced by fungi as propagules during asexual and sexual reproduction (Læssøe and Petersen, 2019). Fungal spores are varied in shape and size, with typical diameters of 1–30 μm, although their diameters can vary from 1–1000 μm (Fröhlich-Nowoisky et al., 2009; Halbwachs and Bässler, 2015; Jones and Harrison, 2004). Spore dispersal is a complex process, affected by size and shape of the spores, meteorological and ecological factors, and the life history of the fungus. Fungi which produce below-ground fruit bodies, and some mycorrhizal fungi without sporocarps, rarely have spores in significant concentrations in the air, and are more commonly spread by animals. Above-ground fungal fruiting bodies with active release mechanisms or tall stipes may release spores higher into the air (Biedermann and Vega, 2020; Dressaire et al., 2016; Halbwachs and Bässler, 2015; Horton, 2017; Kivlin et al., 2014; Lilleskov and Bruns, 2005; Stephens and Rowe, 2020). Spore size, shape, and ornamentation influence their dispersal, with smaller and/or less dense spores able to travel further, although even very large spores may be dispersed by the air (Chaudhary et al., 2020; Norros et al., 2014, 2012). In addition to the spore and fungal characteristics, the meteorology and ecology of the environment both significantly influence spore dispersal (Halbwachs and Bässler, 2015; Oneto et al., 2020). Mature forests generally have complex, multi-layered canopies which affect airflow through the forest, in addition to altering temperature and relative humidity (RH) variability (Bannister et al., 2021; Gilbert and Reynolds, 2005; Norros et al., 2014).

Meteorological factors have also been shown to impact airborne fungal communities significantly. Several studies have measured concentrations of airborne fungal spores in UK (largely in cities), investigating diurnal and seasonal variables as well as meteorological variables on fungal spores. The effects are almost always species-specific, with a broader seasonal trend of fungal spore concentrations being highest in late summer and autumn (Hollins et al., 2004; O'Connor et al., 2014; Sadyś et al., 2016a, 2014). Larger European studies (e.g. Grinn-Gofroń et al., 2019) and reviews have shown that the two most influential meteorological variables for airborne fungal spore concentrations are temperature and RH (Jones and Harrison, 2004; Moore-Landecker, 2011).

As well as being affected on shorter time scales by meteorological factors, airborne fungal concentrations can also be affected by climate change, both due to increases in $CO_2$ concentrations as well as the corresponding increase in temperature, although

effects on fungi may vary depending on the species (Burge, 2002).  Wolf et al. (2003) tested the response of 11 arbuscular mycorrhizal (AM) fungi to $eCO_2$ at the BioCON FACE grassland experiment, with only a single *Glomus* species producing additional spores in the soil. Wolf et al. (2010) demonstrated an increase in spore production from *Alternaria alternata* (a common airborne allergen) under $eCO_2$ in a growth chamber. Similarly, in an Aspen tree (*Populus tremuloides*) open-topped chamber $eCO_2$ experiment, Klironomos et al. (1997) found that airborne fungal spore concentrations increased, which they

suggested was due to corresponding increases in spore concentrations in the leaf litter.

In addition to direct changes in spore production, it is possible that fungal sporocarp production, and therefore spore production, could also increase due to climate change. Ecological measurements show the autumn fungal fruiting seasons in Europe have become longer over the last 50 years. The fungal fruiting season starts earlier and finishes later due to climate change, in addition to more fungi having an additional spring fruiting season (Gange et al., 2007; Kauserud et al., 2012). This climate-

induced phenology change has also been demonstrated in a corresponding increase in the airborne fungal spore season (Sadyś et al., 2016b). These large changes in the fungal season were mainly attributed to temperature increases; however, at the Aspen FACE experiment, it was found that ectomycorrhizal fungi sporocarp production increased under $eCO_2$, so it seems possible that $CO_2$ impacts fungal sporocarp production independently of temperature (Andrew and Lilleskov, 2009). All of these demonstrated changes in fungal phenology, sporocarp production, and sporulation suggest that bioaerosol concentrations

are also likely to change under $eCO_2$. Even if these findings are fungal species-specific, they have potentially wide-ranging effects for forested habitats.

The studies described above demonstrate that fungi are likely to change spore and sporocarp production under $eCO_2$ ranging from 192–600 ppm, in addition to the significant changes witnessed in fungal growing seasons under the current anthropogenic increases and broader climate effects. Even if these findings are fungal species-specific, they indicate that fungal bioaerosols

concentrations can be expected to change under $eCO_2$, with potentially wide-ranging effects for forested habitats (Baird and Pope, 2021) . However, none of the above studies were completed in complex mature woodlands, with the experiments being completed in laboratories, open-topped chambers or, in the case of Aspen FACE, a young plantation forest. Mature and ancient woodlands represent a more complex and diverse environment, which are likely to respond differently to $eCO_2$ than the plantations of young trees on agricultural soil studied in earlier FACE experiments (Norby et al., 2016). There are also few

studies of forest airborne spore concentration responses to $eCO_2$, with the majority of studies focussing on sporocarp production.

In order to study airborne forest fungal bioaerosols under $eCO_2$ in a mature temperate woodland, we installed low-cost Optical Particle Counters (OPCs) into the BIFoR FACE experiment during Autumn 2018. We assume that the bioaerosols can be represented by measured difference between $PM_{10}$ and $PM_1$ mass concentrations, as detailed in the methods section.

Our hypotheses were:

A)  Hourly fungal bioaerosol concentrations will correlate with hourly weather conditions (wind, RH, temperature).

B)  Fungal bioaerosol concentrations will increase in woodland patches treated with $eCO_2$ for two years.

## 2 Methods

### 2.1 BIFoR FACE

The BIFoR FACE experiment is located in a 19.1 hectare mature oak forest in Staffordshire, UK. The overall aim of the FACE experiment is to mimic the effects of anthropogenic climate change by increasing the $CO_2$ concentrations in areas of the forest by 150 ppm above ambient (~400 ppm) (MacKenzie et al., 2020; Norby et al., 2016). This set-up is one of only two FACE experiments in mature forests worldwide and it provides a unique experiment to study mature temperate woodland. In the context of bioaerosols, this experiment allows the investigation of the direct and indirect effects of $eCO_2$ in a minimally

disturbed forest environment (Hart et al., 2019).

As shown in the site map in Figure 1, the $CO_2$ enrichment set-up consists of nine roughly circular "arrays" of three types, each array being approximately 30 m in diameter. Arrays 1, 4, and 6 are fumigated with additional $CO_2$, increasing the atmospheric $CO_2$ by 150 ppm above ambient (to approximately 550 ppm). The $CO_2$, pre-mixed with air, is released into the tree canopies, using pipes running the height of 25 m tall towers around the perimeter of the array. Arrays 2, 3, and 5 have the same tower

set-up, but fumigate only with air. Arrays 7, 8 and 9 are "ghost" arrays without any fumigation infrastructure. Arrays 1–6 are grouped into three treatment pairs, based on pre-fumigation vegetation and soil biochemistry analysis, each with a single elevated $CO_2$ and single ambient $CO_2$ array. The array pairs are as follows: 1 and 3; 2 and 4; 5 and 6. The present study used Arrays 1–6 only, measuring in each pair of arrays consecutively. Fumigation occurs during the oak growing season, approximately 1st April to 31st October during daylight hours.  In 2018, $CO_2$ fumigation ended on the 31st October 2018 at

sundown.

The predominant tree species in the woodland is *Quercus robur* (English oak), with *Corylus avellana* (Hazel) forming a distinct understory layer. Other tree species present include *Acer pseudoplanatus* (Sycamore), *Crataegus monogyna* (Common Hawthorn), and *Ilex spp.* (Holly). A recent vegetation survey, in 2019, was completed to determine the major ground plant species: *Rubus fruticosus* (Bramble)*, Phegopteris connectilis* (Beech fern), a *Hedera sp.* (Ivy), and grasses where the canopy

has been opened for access rides (G. Platt, private communication, 2019). Hanging and fallen deadwood is left in place except where it poses a direct health-and-safety risk.

In order to focus on the fungal bioaerosol component, we took measurements during the autumn, when pollen and non-fungal spore counts from the dominant vegetation were likely to be low. For example, the pollen counts are highest in spring for the two dominant tree species at BIFoR FACE.  UK oak pollen season can range from March to July and lasts for 4–8 weeks, with

peak concentrations usually occurring in May (Grundström et al., 2019). Hazel pollen season falls earlier in the year, starting as early as January, peaking in February or March (National Pollen and Aerobiology Research Unit, 2012). For ground cover species, the grass pollen season peaks in the summer months, sometimes extending into early September (National Pollen and Aerobiology Research Unit, 2012), and *P. connectilis* sporulation peaking mid-July until mid-September (Page, 1997). None of these pollen and fern sporulation seasons coincide with our measurement period. Our measurement period coincides with

the fungal fruiting season at BIFoR (see below), as well as previously measured peaks in UK fungal spore concentrations (Davies et al., 1963).

Monthly macro-fungi surveys were completed during the OPC measurement period. The protocol for the macrofungal survey can be found in Van Norman et al., 2008. These surveys revealed the annual peak of fungal fruiting in 2018 occurred across September and October at BIFoR FACE. Species fruiting in October 2018 included, but were not limited to: *Lactarius quietus*

(oakbug milkcap), at least one *Russula* (Brittlegill) species, *Lycoperdon perlatum* (Common puffball), *Mycena rosea* (Rosy Bonnet) and other *Mycena* species, *Auricularia auricula* (Jelly ear), *Hypholoma fasciculare* (Sulphur tuft), *Xylaria hypoxylon* (Candlesnuff Fungus), and a number of resupinate fungi, including several *Stereum* species. All of these fungi produce spores in the range of 3–9 µm, matching the measurement capabilities of the OPCs (Læssøe and Petersen, 2019). *Auricularia auricula* is an exception; it was observed but produces larger, 16–18 µm-long spores likely outside the OPC size window.

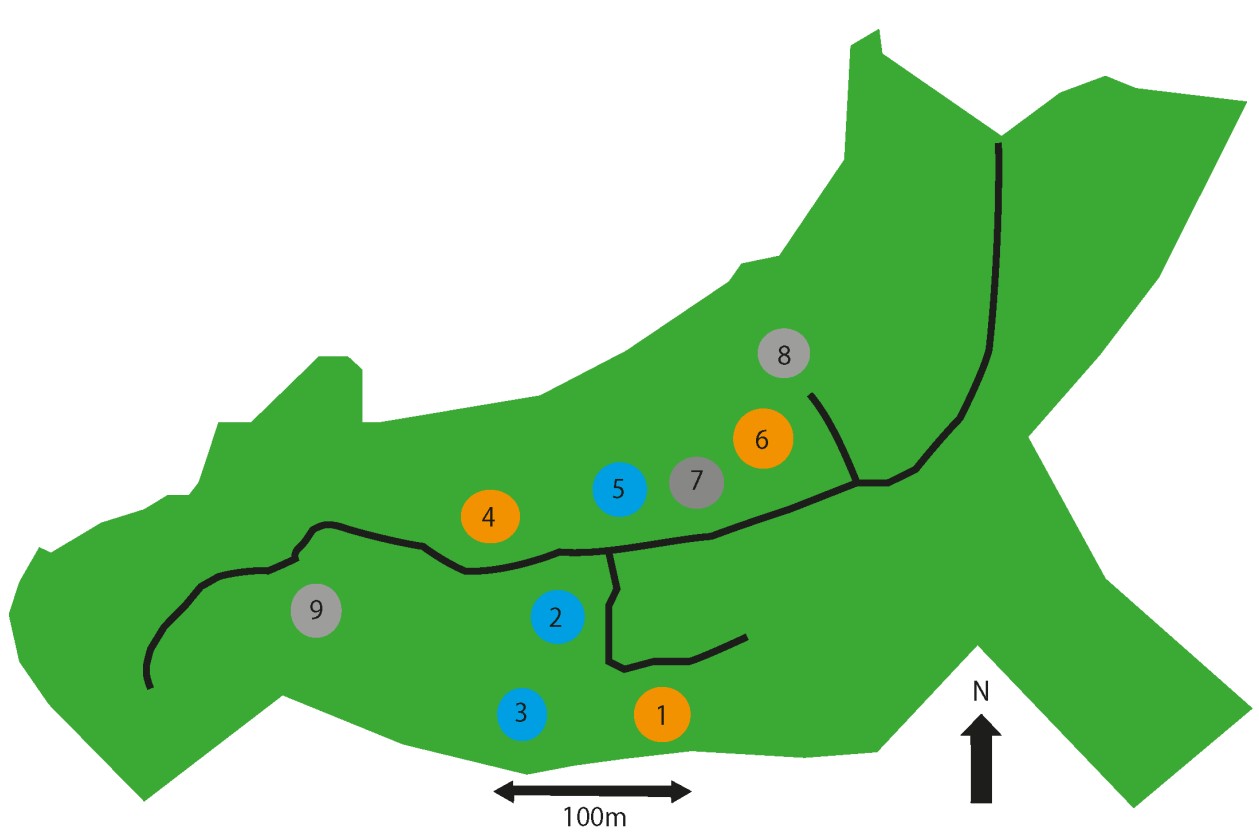


**Figure 1. Map of the BIFoR forest and FACE facility located in Staffordshire (UK) adapted from Hart et al., 2019. The green area shows the whole 19.1 ha Mill Haft woodland, 7.3 ha of which is controlled by the University of Birmingham. The access road is shown as a thick black line. Circles represent scientific research "arrays". Orange arrays 1, 4, and 6 are surrounded with 25 m tall towers. $CO_2$ levels inside the array are raised by fumigation to 150 ppm above ambient (from ~400 ppm to ~550 ppm). Blue arrays**

2, 3, and 5 have similar towers, but are an infrastructure control, spraying out air with current ambient concentrations of $CO_2$. Grey arrays 7, 8, and 9, are non-infrastructure controls, with no towers or fumigation. Treatment and control arrays 1–6 are paired as follows: 1 and 3, 2 and 4, 5 and 6. The landscape surrounding BIFoR FACE is predominantly young tree plantations of mixed broadleaves and conifers (MacKenzie et al., 2020).

## 2.2 Instrumentation

In this study, we use optical particle counters, OPC-N2 sensors (Alphasense, Essex), which count and size particles between 0.3 μm and 10 μm in diameter. The sensors are described in detail by Crilley et al. (2018) and Sousan et al. (2016). In brief, the OPCs count the number of particles, and use particle light scattering to determine particle size using Mie scattering approximations (an de Hulst, 1957). To calculate particle mass concentration, particles are assumed to be of uniform spherical shape, the density is assumed to be 1.65 g cm$^{-3}$ and the refractive index is assumed to be 1.5 + i0. The values for density and

refractive index are the taken from the standard settings of the OPC. The choice of particle density and refractive index has implications for the derived mass concentrations, however comparison between measurements taken within the woodland should be self-consistent It is noted that fungal spores, the target of this study, are often ellipsoidal in shape and can be defined by a short and long axis diameter. Different fungal species generate spores with different shapes, sizes, and density. The assumptions used by the OPC are typically wrong on a single particle (or bioaerosol) basis but should approach reality on an

ensemble averaged basis.

    The sensors do not explicitly discriminate between particle types, so in order to discriminate between fungi and other smaller particles (bacteria and anthropogenic aerosols), we excluded data from particles smaller than 1 μm in diameter, measuring from 1 μm up to the maximum 10 μm measuring capacity of the OPCs. This size discrimination, in conjunction with the experimental location and seasonal timing of the experiment make it highly likely the majority of bioaerosols being captured

are predominantly of fungal origin.

    One of the key concerns raised for the sensing of any aerosol type using low-cost OPCs is that they can report artificially high readings under high RH conditions. Hygroscopic aerosols take up water from the surrounding gaseous environment as a function of RH, with greater water uptake at higher RH. To compare the abundance of aerosol under different environmental conditions, it is preferable to use a 'dry' aerosol concentration, with the hygroscopic growth due to water removed. To

parameterise particle hygroscopicity, κ-Köhler theory is used, which describes the particle hygroscopicity using a single hygroscopicity parameter kappa (κ) (Petters and Kreidenweis, 2007). Pope (2010) details how κ-Köhler theory can be used to link particle mass to hygroscopicity by Eq. (1).

$$a_w = \frac{\left(\frac{m}{m_0} - 1\right)}{\left(\frac{m}{m_0} - 1\right) + \frac{\rho_w \kappa}{\rho_p}} \tag{1}$$

$a_w$ is equilibrium RH expressed as a decimal, $m$ is the wet aerosol mass, $m_0$ is the dry aerosol mass, and $\rho_w$ and $\rho_p$ the density

of water and the dry particulate, respectively. The value of $\kappa$ can be found by curve fitting of the pollen humidograms ($m/m_0$ versus $a_w$).

Biological particles such as fungal spores and pollen have been demonstrated to be hygroscopic, although they typically have a low kappa (κ) value, with pollen κ typically around 0.05–0.1 as opposed to a κ of approximately 0.3 for average anthropogenic aerosols (Griffiths et al., 2012; Pope, 2010). Due to the hygroscopicity of fungal aerosols, high relative humidity can result in a high mass concentration reading in sensors, such as the low-cost OPCs, which do not warm or dry the air. This hygroscopic effect has been explored using low-cost OPCs in urban environments by Crilley et al. 2020 and Crilley et al. 2018, where they used a calibration factor to correct for RH effects.

### 2.3 Instrument installation, bias-checking, and calibration

We used a pair of low-cost OPCs to study bioaerosols at BIFoR FACE. The two OPCs used were controlled to take measurements every 60 seconds using a Raspberry Pi computer (Crilley et al., 2018, based on Hagan, 2017). The OPCs and accompanying instrumentation were installed in a pair of BIFoR FACE arrays at a time (according to the pairing detailed in Section 2.1), between 9th November and 13th December 2018, as per **Error! Reference source not found.**. The air inlet for each Optical Particle Counter was orientated towards the south-west in order to face the predominant wind direction for the site (Hart et al., 2019). The air flows through the instrument at a rate of 5.5 L/min.

Two Tinytag Plus 2 TGP-4500 (Gemini data loggers, Chichester, UK) units were used to measure RH and temperature alongside the low-cost OPCs. The units measure temperature from -25°C to +85°C using an internally mounted thermistor, with the manufacturers stating an accuracy of 0.01°C or better. Under field conditions, the TinyTag sensors closely follow temperature measurements taken by a weather station located at 2 m height in Array 1, with the mean temperature measured by the TinyTags measuring within 0.4°C of the weather station. RH is measured using an externally mounted capacitive sensor, from 0–100% RH, with a manufacturer stated accuracy of +/- 3.0% RH at 25°C. Tinytag units were placed on the top of the OPCs, with the RH sensor facing in the same direction as the OPC inlet.

Above-canopy wind speed was measured using two-D ultrasonic anemometers (WMT700, Vaisala) ~1 m above the canopy (~25 m height) on the northernmost tower of array 1. Below-canopy wind speed and direction (~2 m height) were measured using 03002-L Wind Sentry set (Campbell Scientific, Loughborough, UK) located to the right of the metal walkway near the entrance of Array 1. The manufacturer stated minimum threshold wind speed was 0.8 m s$^{-1}$ for the Campbell anemometer, with 2 m height wind speeds below this minimum threshold considered as zero for the purposes of this experiment.

**Table 1. Table listing the dates (dd.mm.yyyy) and locations of equipment installation across the experimental duration at BIFoR FACE. Both optical particle counters (OPCs) and TinyTag relative humidity and temperature sensors were installed simultaneously.**

| Date range (All dates in 2018) | Tiny Tag unit | OPC unit | Array number | eCO$_2$ or Ambient |
|---|---|---|---|---|
| 9th Nov to 21st Nov | A | 1 | A2 | Ambient |
| | B | 2 | A4 | eCO$_2$ |

| 21st Nov to 30th Nov | A | 1 | A6 | eCO$_2$ |
|---|---|---|---|---|
| | B | 2 | A5 | Ambient |
| 5th Dec to 13th Dec | A | 1 | A1 | eCO$_2$ |
| | B | 2 | A3 | Ambient |


Both the pair of OPCs and TinyTags were installed for a six day side-by-side intercomparison period at 1.5 m height in Array 1 of BIFoR FACE from 30th November 2018 until 05th December 2018. Using the data from this intercomparison week, bias calibration factors were calculated.

No bias factor was applied to the temperature data from the TinyTag units. RH data from the TinyTags was first filtered to

remove any measurements lower than 50% and greater than 99%, and a calibration bias factor of 1.03 applied to TinyTag Unit B to make the two units consistent with each other. A similar calibration was performed on the particulate matter (PM) data collected by the OPCs, with the data from OPC 1 being increased by a bias factor of 1.45. Bias factors were calculated from the six-day side-by-side period, with the factor applied to the data from the full measurement period of 37 days. This bias correction ensured the two OPCs were consistent with each other.

For both the TinyTag and OPC pairs, the consistency between the two instruments, rather than absolute calibration, was more important to allow for observation of differences between two different locations. Hence one instrument of each pair was bias-corrected to the other instrument of the pair. However, a comparison outside of the forest environment was completed between the low-cost OPCs and a TSI 3330 (a reference grade OPC), with the low-cost OPCs measuring within 25% of the TSI without any humidity corrections, which is considered a good performance for the low-cost sensors (Crilley et al., 2020, 2018).

**2.4 Spore dispersal model**

We developed a conceptual model to help interpret the OPC results from BIFoR FACE. Within dense forests, the mean horizontal wind speed $U$ inflects around the tops of the trees, increasing approximately exponentially with height below the inflection and logarithmically with height above it (Bannister et al., 2021; Cionco, 1965; Finnigan, 2000; Raupach et al., 1996). Mean horizontal wind velocity $U$ at each height $z$ inside the canopy can, therefore, be approximated via Eq. (2):

$$U(z) = U_{h_c} e^{-a\left(1 - \frac{z}{h_c}\right)} \tag{2}$$

where $h_c$ is the mean height of the trees and $U_{h_c}$ is $U$ at $h_c$. The attenuation coefficient, $a$, accounts for the flow's response to the density of the forest and tends to increase with increasing leaf area index (LAI) and flexibility of the plant elements (Cionco, 1978; Kaimal and Finnigan, 1994) The exponential relationship in Eq. (2) does not hold in forests with a sparse trunk space and open edges, e.g. many pine plantations, for which a secondary wind velocity maximum occurs below the main crown

(Baldocchi and Meyers, 1988). However, it provides a reasonable first approximation for a forest such as that at the BIFoR FACE site, with extensive understorey growth and closed edges during the leaf-on season.

We adapt the model of Nathan et al., 2002 by using this exponential velocity profile to obtain a rough estimate of the horizontal distance over which spores disperse in a dense forest, $D$. We consider spores evenly distributed within small air parcels within the canopy airspace. $D$ equals the distance an air parcel carrying spores travels between the time of release ($t_0 = 0$) and the time at which the spores in the parcel settle on the ground ($t_1$). As a simplification, here we consider spore 'release' to include both detachment by the mean wind and the point at which spores begin to settle back to the ground after being swept upwards by short intense gusts (Aylor, 1978). For a spore falling at an average settling velocity $v_s$, $t_1 = h_r/v_s$, where $h_r$ is the height of spore release. The vertical position of spores within an air parcel during settling at time $t$ is $z(t) = h_r - v_s t$. Substituting this expression into Eq. (2), we generate Eq. (3):

$$U(t) = U_{h_c} e^{-\frac{a}{h_c}(h_c - h_r + v_s t)} \tag{3}$$

The horizontal distance over which the spores disperse is therefore:

$$D = \int_0^{t_1} U(t)dt \tag{4a}$$

which, taking $t_1 = h_r/v_s$ and $U(t)$ from Eq. (3), gives:

$$D = \frac{U_{h_c} \cdot h_c}{a v_s}\left[e^{-a}\left(e^{\frac{a h_r}{h_c}} - 1\right)\right] \tag{4b}$$

This model assumes the mean wind profile has already adjusted to the presence of the forest, for example, after passing into the forest from surrounding rural areas (see below).

We specified the mean height of the trees as 25 m to reflect those at the BIFoR FACE site. Reported values of $a$ for forests fall within a relatively narrow range of around 1.5–4.8 (Amiro, 1990; Cionco, 1978; Su et al., 1998). We took $a = 2.5$, as measured for forests of similar LAI to BIFoR FACE (Su et al., 1998). Sensitivity testing (not shown) indicated that using higher and lower values of $a$ respectively decreased and increased the occurrence of spores travelling long distances, but had little effect on the modal value of $D$. i.e. using a lower $a$ thickened the right tail of the probability density function of $D$ but affected its peak relatively little.

We performed a stochastic ensemble of model runs ($n$ = 1,000,000) in R (version 4.0.3, http://cran.r-project.org) with low (0–2 m s$^{-1}$), low-medium (*med-l*) (2–3 m s$^{-1}$), high-medium (*med-h*) (3–4 m s$^{-1}$), and high (4–6 m s$^{-1}$) mean wind speeds at the top of the canopy, $U_{h_c}$. We considered only spores in the bottom 10 m of the forest airspace, i.e. $h_r$ in the range 0–10 m. Strong turbulent fluxes of momentum, and scalar quantities such as spores, occur in the upper region of forest canopies (Belcher et al., 2012; Finnigan, 2000), which this simplified model cannot capture. Mean settling velocity, $v_s$, was specified in the range 0.001–0.005 m s$^{-1}$, taking reported values for fungal spores in the size range 1–10 μm (Di-Giovanni et al., 1995; Tesmer and Schnittler, 2007). For each model run, for we specified a random value of $U_{h_c}$, $h_r$ and $v_s$ from a uniform distribution within each of these ranges (using the *runif* function in R).

**2.5 Comparison of measured bioaerosol proxy with regional scale data from Copernicus Atmosphere Monitoring Service (CAMS) global reanalysis (EAC4) data**

To test the assumption that $PM_{10}$-$PM_1$ mass concentration is representative of bioaerosols, the measured $PM_{10}$-$PM_1$ time series is compared with Copernicus Atmosphere Monitoring Service (CAMS) global reanalysis (EAC4) data as a measure of regional aerosol activity influencing the study site. The Copernicus Atmosphere Monitoring Service (CAMS) reanalysis is the latest global reanalysis dataset of atmospheric composition produced by the European Centre for Medium-Range Weather Forecasts (ECMWF), consisting of three-dimensional time-consistent atmospheric composition fields, including aerosols and chemical species(Inness et al., 2019). The nearest available grid point from the EAC4 dataset (longitude = -2.50, latitude = 52.75) to the BIFoR site (longitude = -2.32, latitude = 52.81) and use the same time period as the Ambient and eCO₂ measurements. The $PM_{10}$-$PM_1$ metric is calculated from the EAC4 estimates of $PM_{10}$ and $PM_1$. In addition to the EAC4 $PM_{10}$-$PM_1$ estimate, we also use the aerosol optical depth (AOD) estimates from EAC4, including total AOD, the AOD attributed to dust aerosol, the AOD attributed to sea salt aerosol, and the AOD attributed to sulphate aerosol (Benedetti et al., 2009; Morcrette et al., 2009) . All AODs were for the wavelength of 550 nm.

**2.6 Data processing, visualisation, and analysis**

All data analysis was completed in R version 4.0.3 (R Core Team, 2020), with figures created using openair and ggplot (Carslaw and Ropkins, 2012; Wickham, 2016). Relationships between $PM_{10}$-$PM_1$ concentrations and RH, temperature, and wind speed were visualised used scatter plots and smoothed loess curves, generated in ggplot. Box plots with mean $PM_{10}$-$PM_1$ (and interquartile ranges) were generated to visualise differences in bioaerosol concentrations between eCO₂ and ambient arrays. Scatter plots with regression lines were generated for Figure 7 and for the supplementary figures.

Kruskal-Wallis tests were used to test for statistically significant differences between the eCO₂ and Ambient arrays in Section 3.2.

**3 Results**

**3.1 Hypothesis A: Hourly fungal bioaerosol concentrations will correlate with hourly weather conditions (wind, RH, temperature).**

During the experimental period, there was a total of 46.3 mm of precipitation, with a median temperature of 7.7°C within the forest. The lowest recorded temperature was -1.8°C, and the highest 15.8°C. The average daily temperature peak (and the lowest RH point) occurred at approximately 13:00 local time. RH throughout the measurement period was typically high, with the median and mean measured between 89–91% RH across the duration of the experiment, with the lowest recorded

measurement at 63% RH. Wind speed patterns followed each other closely above and below the canopy, with below canopy measurements significantly lower. The mean wind speed above the canopy was 2.7 m s$^{-1}$, whereas below the canopy it was 0.4 m s$^{-1}$.

**Error! Reference source not found.** Panel A shows the relationship between RH and bioaerosol concentrations at BIFoR FACE: the curve is almost horizontal from 75–90% RH, where at 90% it rapidly increases, most likely indicating the particles becoming exponentially hygroscopic at very high humidities. This sudden increase in particle size at very high humidities demonstrates a clear hygroscopic effect; we therefore performed a correction factor to the data from Crilley et al., 2018, using a $\kappa$ values of either 0.3 or 0.1 (Griffiths et al., 2012; Pope, 2010). Before correction, the median hourly $PM_{10}$–$PM_1$ concentration

under $eCO_2$ conditions was 15.7 $\mu$g m$^{-3}$ and 16.7 $\mu$g m$^{-3}$ under ambient conditions (**Error! Reference source not found.** Panel A). After the application of the Crilley et al. correction factor using a $\kappa$ of 0.3, this decreased to a median of 5.3 $\mu$g m$^{-3}$ under $eCO_2$ and 5.9 $\mu$g m$^{-3}$ at ambient $CO_2$ concentrations (**Error! Reference source not found.** Panel B), and using a $\kappa$ of 0.1, the median was 9.3 $\mu$g m$^{-3}$ under $eCO_2$, and 10.1 $\mu$g m$^{-3}$ under ambient conditions (**Error! Reference source not found.** Panel C). The ratio between the mean concentrations obtained from the $eCO_2$ and ambient plots depend upon whether the Crilley

correction factor is used and what value is used for $\kappa$. The values are 94%, 90%, and 92% for the uncorrected ratio, the ratio using $\kappa = 0.3$ and the ratio using $\kappa = 0.1$, respectively. A comparison of the data before and after correction is shown in **Error! Reference source not found.** panels D, E, and F. Due to the decrease in PM concentrations, with respect to RH, shown using a $\kappa$ of 0.3, the data corrected using a $\kappa$ of 0.1 was used for all further analyses, as this was deemed more appropriate given the likely particle composition (low hygroscopicity bioaerosols). After correction to remove the instrument artefacts, there was no

clear effect of RH on bioaerosol concentrations. If the observed median concentrations of $PM_{10}$-$PM_1$, using the Crilley correction with a $\kappa$ of 0.1, were solely composed of idealized spherical fungal spores with radius of 3 $\mu$m and density of 1.65 g/ml this would equate to spore concentration of approximately 50,000 spores m$^{-3}$, which falls at the high end of Sesartic and Dallafior (2011b) estimate of ground level spore concentrations of 10,000-50,000 m$^{-3}$.

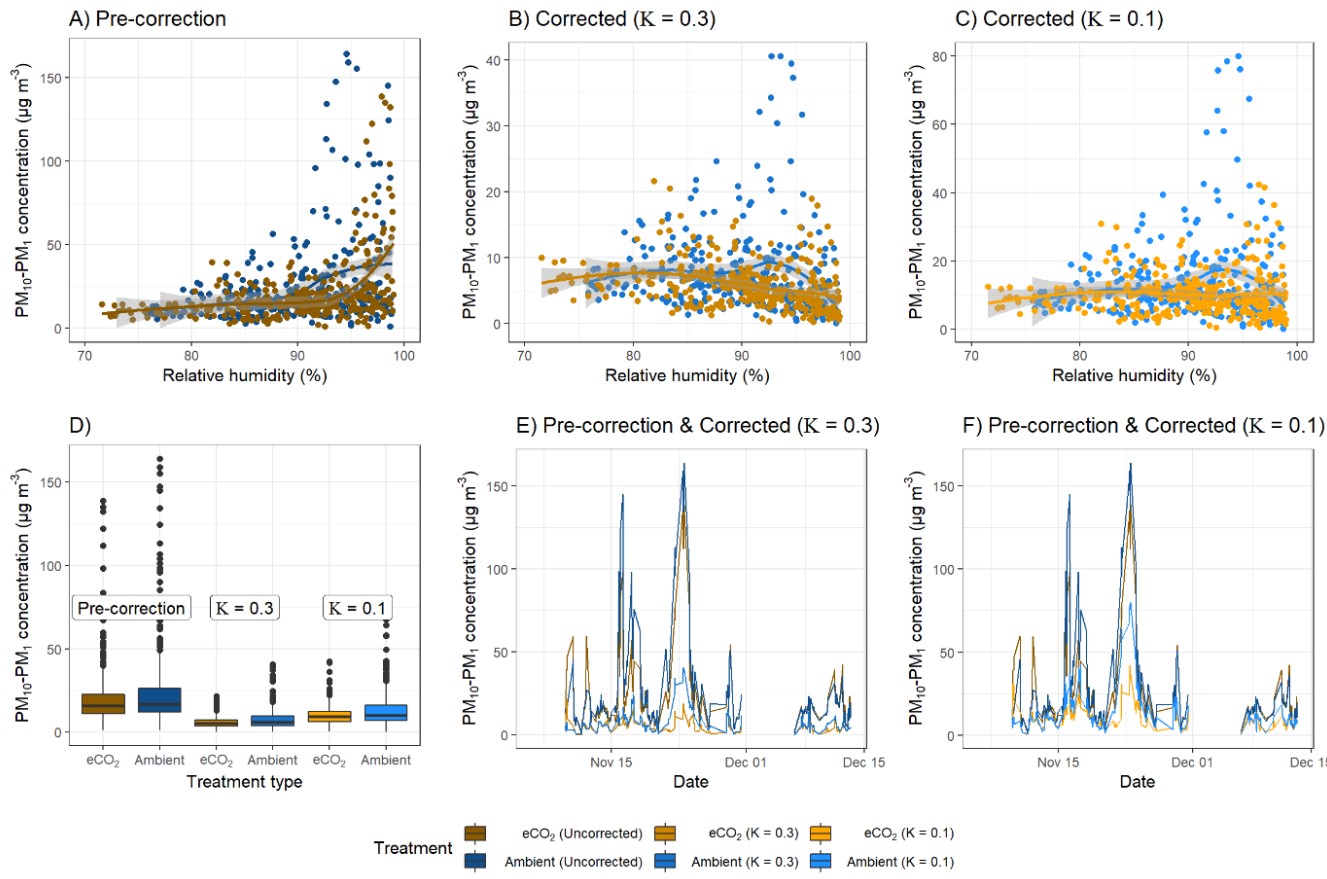

*Figure 2. Relative humidity and bioaerosol concentrations (PM$_{10}$–PM$_{1}$) at BIFoR FACE in Autumn 2018. Panel A shows PM$_{10}$–PM$_{1}$ concentrations with increasing RH without any correction applied. Panels B and C plots the same data as Panel A, however a correction factor from Crilley et al., 2018 has been applied to the bioaerosols data to correct for the hygroscopic effect of the particles, Panel B using a κ of 0.3, and Panel C a κ of 0.1. Panels A, B, and C, all have loess curves fitted (Wickham et al., 2019). Panels D, E, and F compare the data before and after corrections. All the plots show PM$_{10}$–PM$_{1}$ concentrations under ambient conditions (Arrays 2, 3, and 5) in blue colours, and eCO$_2$ conditions (Arrays 1, 4, and 6) in orange colours. Data pre-correction is shown is dark orange and blue, corrected data using a κ of 0.3 mid-blue and mid-orange, and the corrected data using a κ of 0.1 uses the lightest blue and orange. All panels use hourly averages of PM$_{10}$-PM$_{1}$ concentrations.*

The effects of wind speed and direction on the bioaerosol concentrations were also investigated in depth. Anemometers were located both above (25 m height) and below (2 m height) the canopy, with below-canopy measurements broadly following those above the canopy but measuring significantly lower at the lower height (**Error! Reference source not found.** Panel A). Above the canopy, the hourly wind speed never dropped below 0.8 m s$^{-1}$, whereas recorded speeds of 0 m s$^{-1}$ (i.e. below

anemometer threshold) were common below canopy. The maximum hourly wind speed above canopy was 7.05 m s$^{-1}$, compared with 2.59 m s$^{-1}$ below canopy.

Bioaerosol concentrations demonstrated high variability under changing wind speeds; however, a possible U-shaped curve was observed, with higher bioaerosol concentrations at the lowest and highest wind speeds measured, as shown in **Error!**

**Reference source not found.** Panels B and C. This effect was shown using the wind speeds taken from both above (25m) and below (2m) the canopy. Figure 4 shows the effect of wind speed and direction on bioaerosol concentrations using *openair* polar bivariate plots (Carslaw and Ropkins, 2012). Observed wind speed patterns follow broadly similar trends between each pair of arrays, with the majority of medium concentration (medium concentration (green-yellow) bioaerosol activity occurring in the SW and SE quadrants, and only low (blue) concentrations under northerly winds. Peaks in bioaerosol concentration

(presumed high sporulation events) are visible in red, with some events being replicated across both eCO$_2$ treatment and control (e.g. the SW quadrant event in A5 and A6), and other high PM events only occurring in a single array (e.g. the SE quadrant event in A4). By detecting high PM events in a single array at a distinct time shows that the OPCs can detect differences between the BIFoR FACE arrays.

To investigate the relationship between spore dispersal and wind speed further, we used the model outlined in Section 2.4 to

investigate the horizontal distance over which spores disperse, $D$. The model was run (n = 1,000,000) for spore release heights ($h_r$) of 2 m, 5 m, and 10 m. Figure 5 shows the probability density functions of $D$ under four wind speed scenarios: low, low-medium, medium-high, and high. As would be expected, generally, the lower the spore release height and wind speed, the shorter the overall distance travelled by spores. Most wind conditions experienced at BIFoR FACE fall under the "low" wind speed scenario (black lines), for which the modal distance travelled by spores were less than 20 m for all release heights (Figure

5). With the arrays having diameters of 25–30 m, this means that under typical conditions, we could expect spores to stay within the array they were released in or, at least, very unlikely to be transported into neighbouring arrays (i.e. a spore released under eCO$_2$ is unlikely to be measurable in an ambient array in low wind conditions). However, at higher wind speeds, a heavy tail is present on each of the plots, indicating that spores are more likely to travel much greater distances at higher wind speeds, potentially causing mixing between the arrays. Finally, we looked at PM$_{10}$-PM$_1$ concentrations under changing temperature

(Figure 6). There was a small linear positive relationship between temperature and bioaerosol concentrations, with PM$_{10}$-PM$_1$ concentrations of around 5 µg m$^{-3}$ at the lowest temperatures (1-3 °C), increasing up to 10 µg m$^{-3}$ at the highest temperatures of 12-13 °C.


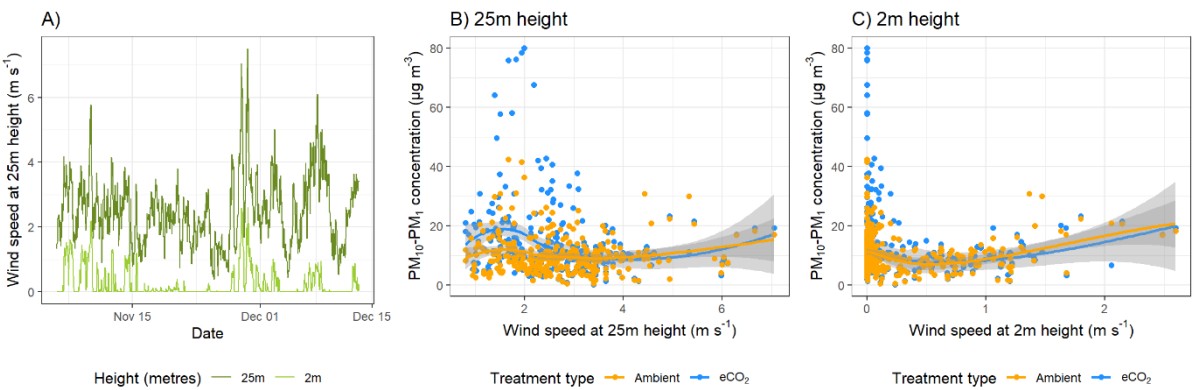

*Figure 3. Wind speed and bioaerosol concentrations at BIFoR FACE in Autumn 2018. Panel A shows the hourly average*
       *wind speeds above (25 m height) and below (2 m height) the canopy across the experimental duration. Wind speed data at*
       *25 m height is in dark green, and data from 2 m height in light green. Panels B compares hourly average $PM_{10}$–$PM_1$*
       *concentrations with hourly average wind speeds measured at 25 m height, and Panel C the same $PM_{10}$–$PM_1$ concentrations*
       *using wind speeds measured at 2 m height. Panels B and C both have loess lines of fit (Wickham et al., 2019), with data from*
*ambient arrays (Arrays 2, 3, and 5) in blue, and data from $eCO_2$ arrays (Arrays 1,4, and 6) in orange.*

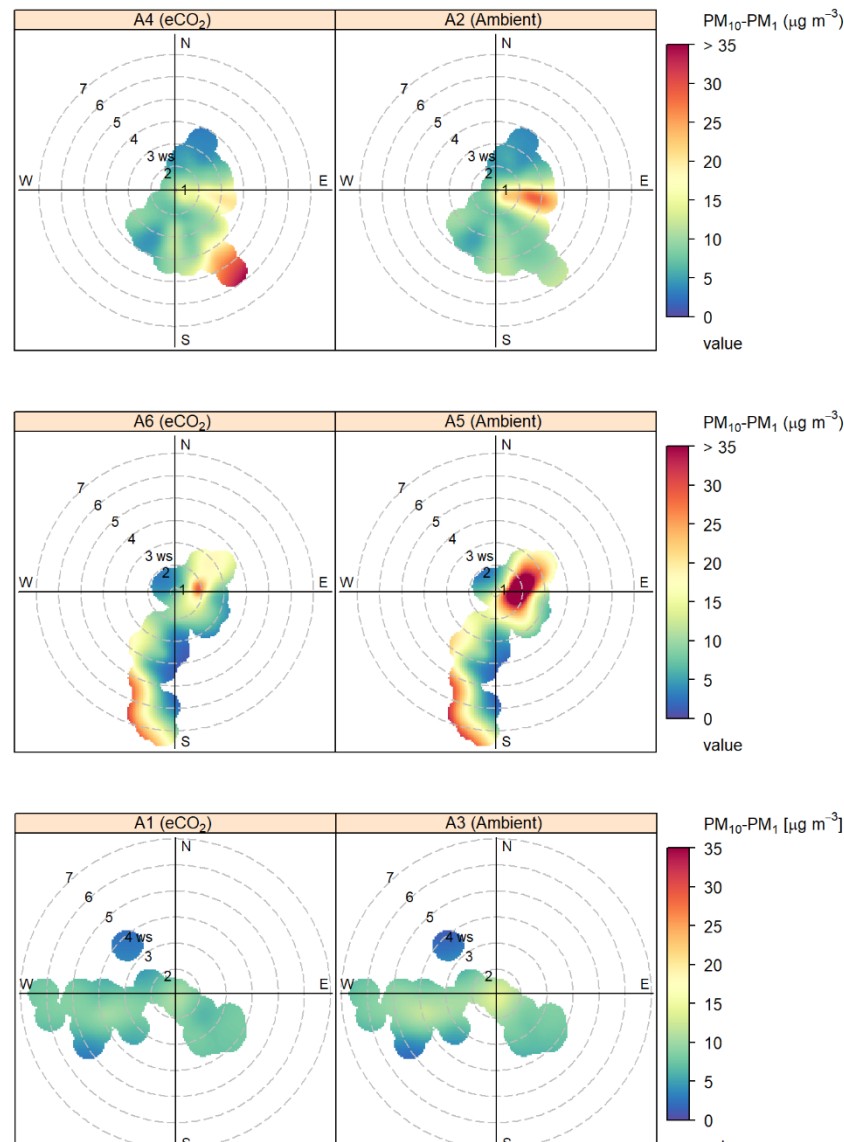

**Figure 4. Openair polar bivariate plots showing bioaerosol concentrations ($PM_{10}$-$PM_1$) with wind speed and direction across the Autumn 2018 experimental duration at BIFoR FACE (Carslaw and Ropkins, 2012). Plots are displayed in the three array pairs (one eCO₂, one ambient) in which the pair of OPCs were located. Dates of OPC installation in each array pair are detailed in** Error! Reference source not found.**. Bioaerosols were measuring using the OPCs at 2 m height, and wind data was taken from the anemometers at 25 m height. Colour gradients display the concentrations of bioaerosols detected, with low concentrations shown in blue/green colours, mid concentrations in yellow, and high concentrations in orange/red.**

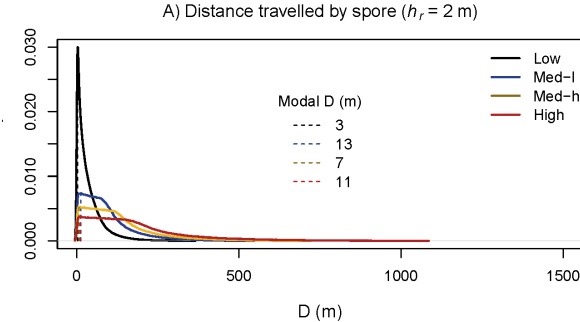

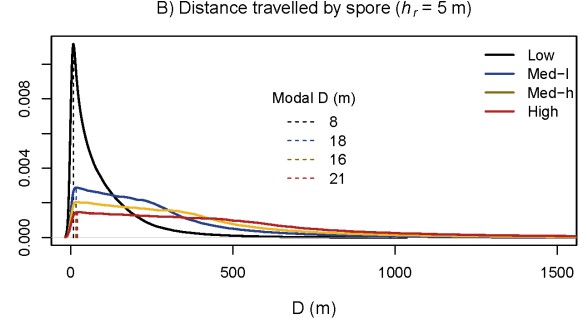

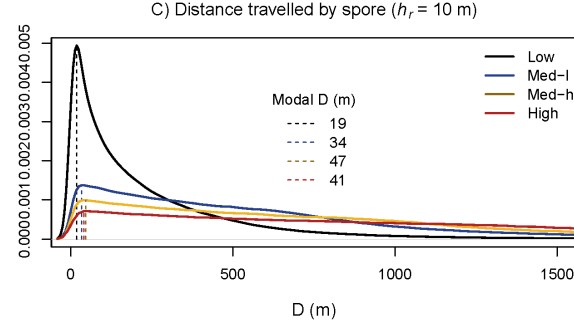

**Figure 5. Model outputs of probability density functions for distance travelled by spores in a forested environment. Panels A, B, and** 385 **C had different spore release heights (hr) inputted into the model- 2, 5, and 10 m respectively. Solid lines are in different colours for each of the low (black), low-medium(blue), medium-high(yellow) and high (red) $U_{h_c}$ (wind speed) cases. Dashed lines show modal D (distance) travelled by the spores. n = 1,000,000 for each case.**

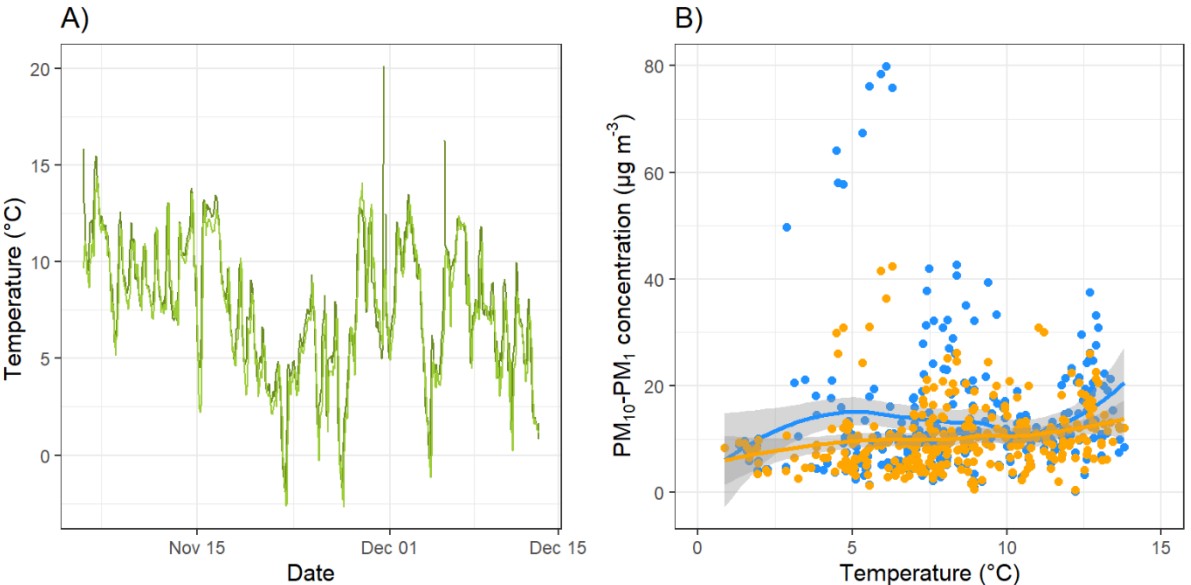

Figure 6. Temperature and bioaerosol ($PM_{10}$-$PM_1$) concentrations across the experimental duration at BIFoR FACE. Panel A shows temperature data, with mean temperature measured by TinyTag sensors shown in dark green, and light green showing data collected by the Array 1 weather station. Panel B compares temperature data with bioaerosol concentrations, with data from ambient arrays (Arrays 2,3,and 5) in blue, and data from $eCO_2$ arrays (Arrays 1,4, and 6) in orange, with both treatment types fitted with a loess line (Wickham et al., 2019).

**3.2 Hypothesis B: Fungal bioaerosol concentrations will increase in woodland patches treated with $eCO_2$ for two years.**

Figure 7 shows six plots comparing $PM_{10}$-$PM_1$ concentrations between the $eCO_2$ (orange) and ambient (blue) arrays. When comparing total concentrations across the entire measurement period (Figure 7 Panels A, B, and C), there was no significant difference (p = 0.489) between the $eCO_2$ treatment and ambient, however a heavy tail is present in the ambient arrays. To investigate this relationship further, we separated out the data into low bioaerosol concentration conditions (<10 µg m$^{-3}$) and high bioaerosol concentrations (>= 10 µg m$^{-3}$) (Figure 7 Panels D, E, and F). When bioaerosol concentration are low (Figure 7**Error! Reference source not found.** Panels D and E), there was no significant difference between the $eCO_2$ and ambient treatments (p = 0.689). These low concentrations likely represent background levels of aerosols that are consistent throughout the forest. However, when selecting for high concentration events, there was a significant effect seen, whereby $eCO_2$ treatment suppressed concentrations of high bioaerosol events (p = 0.023) (Figure 7 Panels D and F).

The boxplot in Panel A shows that there is no significant difference between the medians of $eCO_2$ treatment versus the control; however, there were slightly higher bioaerosol concentrations shown under ambient conditions. Panel B shows a time series of the data, demonstrating that bioaerosol concentrations match extremely closely between the two treatment groups. The small difference in the overall medians can be largely attributed to the $PM_{10}$-$PM_1$ concentration difference between the $eCO_2$ and

ambient during the largest bioaerosol event around the 22nd of November (during the second measurement period, shown shaded in grey), where the ambient array (shown in blue) measures higher than the eCO2 array in orange.

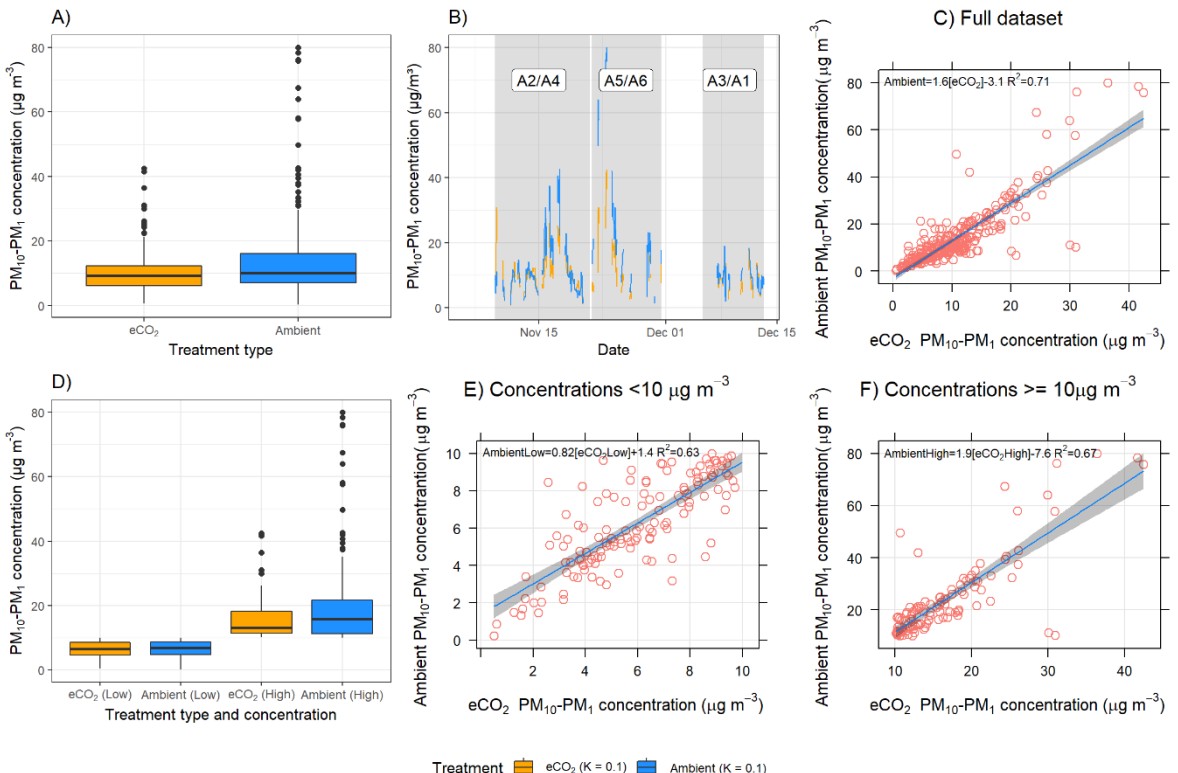

**Figure 7. Bioaerosol concentrations across the experimental duration in ambient (Arrays 2,3, and 5) and eCO$_2$ (Arrays 1, 4, and 6) arrays. eCO$_2$ treatment is shown in orange, ambient control in blue. Panels A, B, and C include the full dataset, with no significant difference shown between the eCO2 treatment and control. Panels D, E, and F split the dataset into low (<10 µg m$^{-3}$) and high (>=10 µg m$^{-3}$) bioaerosol concentrations. When the data is split in this way, there was no significant difference between treatments under low bioaerosol concentrations. However, when bioaerosol concentrations were high, concentrations were significantly lower under eCO$_2$ treatment.**

### 3.3 Comparison of the measured bioaerosol proxy with regional scale data from Copernicus Atmosphere Monitoring Service (CAMS) global reanalysis (EAC4) data

It can be seen in Figure 8 that the EAC4 estimate of regional $PM_{10}$-$PM_1$ does not correlate with the $PM_{10}$-$PM_1$ measured within the forest. Hence, we believe the measured $PM_{10}$-$PM_1$ is local to woodland and not representative of the regional air mass. The Pearson correlation between the measured $PM_{10}$-$PM_1$ in the Ambient and eCO$_2$ arrays and the EAC4 estimates is -3 and 3, respectively, highlighting negligible correlation between the measured and EAC4 timeseries. To look for further possible correlations between the measured $PM_{10}$-$PM_1$ and estimated aerosol properties, correlative analysis was conducted on the measured $PM_{10}$-$PM_1$ with the aerosol optical depth (AOD) estimates from EAC4 consisting of the total AOD, the AOD attributed to dust aerosol, the AOD attributed to sea salt aerosol, and the AOD attributed to sulphate aerosol. Similar to the

EAC4 PM$_{10}$-PM$_1$ product, there were no non-negligible correlations with the AOD products, with Pearson correlation values ranging from -16 to 23. See supplementary material for more information the correlation plots provided in Figures S1 to S11.

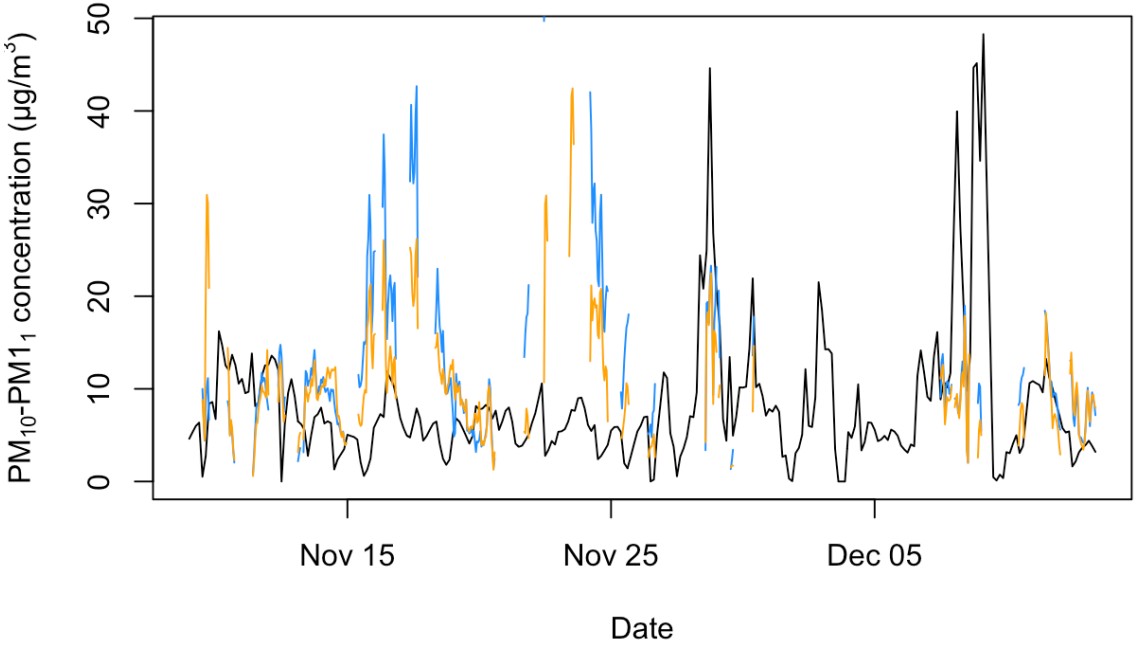

**Figure 8. Time series data of the PM$_{10}$-PM$_1$ mass concentrations measured in the Ambient and eCO2 arrays and provided by the Copernicus Atmosphere Monitoring Service (CAMS) global reanalysis (EAC4) dataset. The BIFoR data contains only data collected when RH < 99%.**

## 4. Discussion

In support of hypothesis A, we found that various meteorological variables affected bioaerosol concentrations. The RH before correction was of particular importance, especially as the low-cost OPCs do not have any warming or drying capacity, and therefore are susceptible to recording higher mass concentrations of aerosols under high humidity conditions as particles swell
with additional water (Crilley et al., 2018). We found that the RH % threshold for significant particle swelling was 90–95%, which is a much higher RH value than would be expected for anthropogenic aerosols, which typically contain more hygroscopic components including salts, and therefore provides evidence that the measured PM$_{10}$–PM$_1$ fraction represents a predominantly biological source. In their study investigating effects of RH on fungal spore swelling, Reponen et al. (1996) demonstrated a similar effect whereby a significant swelling of fungal spores was seen, but only at very high humidities (greater

than 90%). We believe this threshold for particle swelling further demonstrates that we are recording a biological source. This hygroscopic evidence is in addition to the ecological and phenological evidence for spores being the dominant source within the $PM_{10}$-$PM_1$size fraction during the measurement period.

After correction, we did not see any effect of RH on fungal spore concentrations. The evidence in the literature for the relationship between spore concentrations and RH is mixed, probably largely due to species-specific effect. Some fungal species are dependent on high RH to develop spores (and then release spores by rain droplets), whereas other species release spores in response to a drop in RH (Jones and Harrison, 2004; Li and Kendrick, 1995). The high diversity of fungal species at BIFoR (each with possibly varying responses to RH) could result in this flat line. Another possible explanation is that the fungi never experience low RH conditions due to the constant high RH at BIFoR FACE, potentially with the RH never dropping low enough to trigger a burst of spore release. Finally, it is important to mention that although the Crilley correction is an essential part of using low-cost sensors in a high-humidity environment, the high swelling threshold of fungal spores means that the κ value of 0.3 (or even 0.1) may be inappropriately high for biological particles such as these. As shown in **Error! Reference source not found.** Panel B and C, after correcting the data, the $PM_{10}$–$PM_1$ concentrations decrease after 95% RH, although this effect is smaller when using a κ of 0.1. For future studies, calibrating the low-cost OPCs against a reference grade instrument in the experimental environment in order to provide a calibrated κ-value would be an appropriate measure, and enable a more thorough investigation of the RH dependence, and improve the accuracy of the Crilley correction for bioaerosols (Crilley et al., 2020).

We measured an approximately U-shaped curve of bioaerosol concentrations in response to wind speed, with the highest $PM_{10}$–$PM_1$ concentrations being displayed at low and high wind speeds. This likely indicates a balance between spore release mechanisms and mixing of low-aerosol air. At low wind speeds, there is minimal movement of air through the forest, so any spores released do not travel significant distances. At high wind speeds, there is much higher movement of air, which decreases concentrations by carrying spores further; however, spore release by wind could also be increased (Dressaire et al., 2016). At medium wind speeds, a balance between these two effects occurs, maintaining bioaerosol concentrations at a lower level (Jones and Harrison, 2004).

Our model demonstrated that under low wind-speed conditions, spores are most likely to travel a relatively short distance, suggesting that between-array mixing was unlikely under the meteorological conditions experienced during the measurement period. Particle release height ($h_r$) had a significant impact on the distance travelled by a particle, which is an important consideration given the source of bioaerosols. For example, fungal spores are likely to have a significantly lower $h_r$ than pollen which, based on our model, would significantly impact the distance travelled by the different bioaerosol types. Due to the complexity of the forest environment, there are a number of aspects which the model did not capture which could alter the distance travelled by a spore. Complex wind dynamics such as ejections (infrequent but strong upward gusts of air) and turbulence around forest edges could increase the distance travelled by a spore, as well as varied spore release dynamics between fungi (Dressaire et al., 2016; Norros et al., 2014). The model also assumes a spatially representative wind speed profile, which does not account for local velocity effects induced by gaps, clearings, and changes in canopy density. These

local effects may be important near the ground, where wind speeds are generally low. However, the distance travelled could
also be shorter than modelled, for example, the model does not capture spore deposition onto forest surfaces other than the ground, spores being swept downwards by gusts, and changes with wet versus dry deposition.

The data demonstrated a decrease in bioaerosol concentrations at lower temperatures, which is the expected response within the range of temperatures we observed in a woodland of this type (Gange et al., 2007). The number of fungal sporocarps present across the duration of the experiment decreased, and it is therefore logical to expect that spore production would also
consequently decrease. For future studies, it would be interesting to begin measurements earlier in the fungal season, allowing us to capture the peak of sporocarp production, as well as the end of the active $eCO_2$ fumigation at the FACE experiment.

Regarding Hypothesis B, to our knowledge, this is the first assessment of bioaerosols in any forest FACE experiment to date, and therefore provides valuable data contributing to the understanding of the forest environment at BIFoR FACE as well as more broadly in the context of forests, and FACE experiments. We did not see a significant effect of $eCO_2$ on total aerosol
concentrations. However, when data was split into low and high aerosol concentrations, two differing responses were demonstrated. Under low concentrations, there was no significant effect of $eCO_2$ treatment, we therefore think these low concentrations represent the background concentrations of aerosols present both inside and in the wider environment around the forest. However, when high concentrations of bioaerosols were detected, $eCO_2$ treatment resulted in a suppression of $PM_{10}$-$PM_1$ concentrations. This is demonstrated in a number of larger sporulation events during the experimental period, for example
the large sporulation event peaking on $22^{nd}$ November, where although both arrays showed the sporulation event, the magnitude of this event was significantly different between $eCO_2$ and ambient.

This evidence, as well as the $eCO_2$ dispersal data from Hart et al., 2019, and the modal spore travel distances shown in our model suggests to us that this experimental set-up is capable of detecting differences between the $eCO_2$ and ambient arrays. It
is apparent that at background concentrations these differences are not present between arrays, however there may be a response to $eCO_2$ during large sporulation events. This could be for a number of reasons, the primary reason being that the experiment was in the very early stages of fumigation (year two of ten total years planned), and there might be a lag expected in fungal responses, as they are most likely responding indirectly to $eCO_2$ via changes in leaf litter and soil. Although the literature states a small number of cases of where individual fungal species do respond directly to $eCO_2$, it is likely that the
main effects would be secondary, for example, competition between autotrophs and heterotrophs for nutrients, or an increase in leaf litter production resulting in an increase the population of decomposer fungi. There is evidence for increased autotrophic productivity under $eCO_2$ at BIFoR FACE in the form of increased leaf-scale photosynthesis (Gardner et al., 2020) and fine root production (Ziegler et al., 2021). Whether this increased autotrophic productivity primes, or competes with, fungal activity requires further work. Continuing to monitor bioaerosol concentrations throughout the 10+ year experimental duration of the
BIFoR FACE experiment, along with monitoring in other FACE experiments, will be key in understanding how $eCO_2$ affects bioaerosol concentrations long term.

Another possible reason for not picking up an $eCO_2$ response in fungal bioaerosols in the full dataset is that the maximum diameter of the particles that can be detected by the OPCs is 10 μm, which does exclude several fungal species, including those known to be present in the BIFoR forest during the experimental duration. However, we note that many common woodland spore species are smaller than 10 μm, including the following species observed in the forest: *Lactarius quietus*, *Russula* species, *Lycoperdon perlatum*, *Mycena rosea* other *Mycena* species, *Auricularia auricula*, *Hypholoma fasciculare*, *Xylaria hypoxylon*, and various *Stereum* species. In addition to the observed species, many spores commonly observed to be airborne in the UK, including *Cladosporium*, *Ganoderma*, and *Aspergillus* species (Sadyś et al., 2016), have diameters less than 10 μm. If a response to $eCO_2$ is species-specific, then it is possible that we are missing an effect in fungi with larger spores. It is also possible that although the total and background aerosol concentration was stable under $eCO_2$, the aerosol composition could have been different (e.g. altered ratios of fungal species present), which we were not able to detect using the OPCs. Varying fungal species between the ambient and $eCO_2$ arrays could therefore be responsible for the differing response we detected during sporulation events. In order to determine definitive particle composition, other techniques such as detectors using fluorescence, or DNA sequencing of biological material is required (Fröhlich-Nowoisky et al., 2016; Gosselin et al., 2016; Healy et al., 2012). Finally, our experimental measurement period occurred outside of the $eCO_2$ fumigation season, and therefore the main growing season of the trees, which could have reduced any direct effects of the $eCO_2$, although, given the likely cumulative effect of the $eCO_2$ treatment over the growing season, and the fact that our measurements were focused on the period of peak sporulation, it seems more likely that our measurements were well-timed to observe any treatment effects.

**5 Conclusions**

We have demonstrated that low-cost OPCs are suitable for measuring $PM_{10}$-$PM_1$ concentrations in forests, or other high-humidity environments. We demonstrate that the $PM_{10}$-$PM_1$ metric is a good proxy for bioaerosols because of the bioaerosol representative size range, the location of the study site (a woodland in a rural location), the field measurement taking place during the season of peak fungal activity, and the low hygroscopicity of the particles measured. Through comparison with the EAC4 estimate of $PM_{10}$-$PM_1$ we highlight that the woodland measurements do not follow regional air pollution trends and that the observed $PM_{10}$-$PM_1$ concentrations are likely from woodland sources. The findings from this study have consequences for other research into the aerobiology of forests, and also opens up the bioaerosol research field to a wider array of locations and researchers. The low-cost sensors measured significant swelling in $PM_{10}$–$PM_1$ sized particles at very high RH, which we corrected for using a calibration factor from Crilley et al. (2018). For future work, generating a κ value using a reference-grade instrument in situ would improve accuracy (Crilley et al., 2020). Temperature, wind speed, and wind direction were all shown to affect bioaerosol concentrations; however, we did not see any effect of RH (post-correction). Treatment with $eCO_2$ may repress concentrations of bioaerosols during high sporulation events (p = 0.023), however this significant difference was not detected across all concentrations of aerosols (p = 0.489). Therefore, further investigation later in the 10+ year experimental duration is warranted, as well as investigating forest bioaerosols in other forest FACE experiments globally. The use of the

PM$_{10}$-PM$_1$ metric as a proxy for bioaerosols, in woodland and other settings, should be further evaluated through future experiments that unambiguously measure bioaerosol concentrations.

## 6 Data availability

Data supporting this publication are openly available from the UBIRA eData repository at https://doi.org/10.25500/edata.bham.00000688. The data provided is the hourly data with the inter-unit calibration bias factor applied.

## 7 Author contribution

ABB and FDP designed the bioaerosol study as part of the FACE programme designed by ARMK. ABB collected and analysed the OPC data and prepared the manuscript. EJB wrote the model. All co-authors discussed the results and contributed to writing the manuscript.

## 8 Competing interests

The authors declare that they have no conflicts of interest.

## 9 Acknowledgements

The BIFoR FACE facility is a research infrastructure project supported by the JABBS Foundation and the University of Birmingham. ABB is supported by a Natural Environment Research Council (NERC) studentship through the DREAM programme (grant NE/M009009/1), EJB is supported by a NERC studentship through the CENTA programme (grant NE/L002493/1). ARMK gratefully acknowledges support from NERC through grants NE/S015833/1 and NE/S002189/1. FDP gratefully acknowledges support from EPSRC through grant EP/T030100/1. With thanks to the British Pteridological Society for their literature recommendations and advice on fern phenology.

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
