# Peer review of "Mass concentration measurements of autumn bioaerosol using low cost sensors in a mature temperate woodland Free Air Carbon Dioxide Enrichment (FACE) experiment: investigating the role of meteorology and carbon dioxide levels"

_Biogeosciences, 2021_

## Author Comment (AC1)

Thank you for your comments and feedback on "Mass concentrations of autumn bioaerosol in a mature temperate woodland Free Air Carbon Dioxide Enrichment (FACE) experiment: investigating the role of meteorology and carbon dioxide levels". We address each of your comments in turn below.

Firstly, you highlight correctly that the optical particle counters (OPCs) do not explicitly discriminate between bioaerosols and other aerosol compositions. It is correct that the OPCs cannot discriminate between particle types (as stated in section 2.2 of the paper). However, we believe that due to the location of the study site (a woodland in a rural location), the field measurements dates in peak fungal activity, and the low hygroscopicity of the particles we measured, it is very likely we are detecting a predominantly biological source. You suggest there will be a notable amount of sand particles or plant debris present. We include plant debris in our definition of bioaerosols, as stated in the abstract, although it is likely that fungal spores will dominate the bioaerosol fraction due to the timing of the experimental duration. For the measurement period of the experiment, it is unlikely there are any great intrusions of sand particulates. The UK does experience desert dust inclusions, especially in the Spring but less so in the autumn. During the field campaign period, comparison with the outputs from CAMS global reanalysis (EAC4) https://www.ecmwf.int/en/forecasts/dataset/cams-global-reanalysis Dust aerosol optical depth at 550 nm data product shows no correlation with the measured OPC PM10-PM1 data from this paper. We agree that it would be advantageous to have aerosol chemical composition measurements in addition to aerosol size measurements, but practical constraints made that impossible in this field campaign. We hope that future field campaigns will allow for more extensive equipment payloads to be utilized.

The OPCs used are only capable of measuring particles up to 10 µm in size, and therefore some larger fungal spores and pollen particles are not capable of being detected. We explicitly highlight this limitation on line 437 of the present manuscript. However, many common woodland spore species are smaller than 10 µm, including the species listed on lines 126-130 of the paper, as well as several species of airborne spores shown to be extremely common across the UK including Cladosporium, Ganoderma, and Aspergillus (Sadyś et al., 2016).

We aim to show that the benefits of the low-cost OPC outweigh some of the negatives (such as the limited size range) because they enable more studies to be undertaken on forest bioaerosols. We recognise that low-cost OPCs are not sufficient to classify and categorise bioaerosol unambiguously (as stated on line 151), but we believe we have shown that — with due caution not to over-interpret the data — OPC data can yield meaningful data addressing significant science questions. We note that this is the first study on bioaerosols (or aerosols) in any of the current or previous forest FACE experiments.

You note the importance of the relationship between RH and particle counts. We have discussed this extensively in the paper in sections 2.2, 3.1, and in the first two paragraphs of the discussion, including citing the relevant literature on hygroscopic corrections of data from OPCs. One signature of bioaerosols is their low hygroscopicity, which is observed in this study. In the conclusions we note that additional work in characterising the kappa values required for bioaerosols would be useful future work.

We describe the instrument set-up in Section 2.3, including the OPC specifications, experimental duration, and the height at which the instruments were installed.

Regarding high-resolution wind measurements and spore dispersion; this is a good idea but was beyond the scope of this paper. High-resolution 3D observations, from which the turbulence kinetic energy can be calculated, are available around the edge of the present site. However, a high density of equipment would be required to obtain spatially representative 3D velocity measurements in each study array. We hope that future field campaigns will allow for more extensive equipment payloads to be utilized, for example, to investigate the response of spore concentrations to gusts. Because of the difficulties inherent in interpreting 3D velocities near forest floors—i.e. those most relevant to woodland fungi— in this study, we do not rely on the high-resolution measurements from around the edge of the site.

Regarding the effects of $eCO_2$ on bioaerosol concentrations, we have discussed the possible direct and indirect effects in the final two paragraphs of the discussion (section 4) including changing fungi speciation, changing habitat and fungal substrates. This discussion includes prior work on the effect of $CO_2$ on fungi. This is the first study on aerosols or bioaerosols in a forest FACE experiment, and we hope the greater accessibility of low-cost sensors will enable more bioaerosol studies in FACE experiments in the future, providing further evidence on the direct and indirect effects of $eCO_2$. We believe this paper is an important first step to being able to understand the role of the changing $CO_2$ levels upon atmospheric bioaerosol concentrations.

70% of measurements occurred during low, and low-medium wind speed conditions (less than 3 m s$^{-1}$), whereby the model demonstrated that mixing between arrays is highly unlikely. Even at these lower wind speeds, the effects of $eCO_2$ discussed elsewhere in the paper still apply- whereby high concentration events are suppressed by $eCO_2$, but lower background concentrations are the same between $eCO_2$ treatment and control.

---

## Author Response (AR1)

Author's Response

Dear Editor and Reviewers many thanks for your careful reading of our manuscript and your helpful suggestions. Please find our responses to your points below. We cluster similar questions from reviewers/editor together. In addition to the reviewers comments we have tidied up the manuscript in certain places (typos, too long sentences etc.).

Comments by reviewer Branko Sikoparija (BK)
Comments by reviewer Matt Smith (MS)
Comments by reviewer Madeleine Petersson Sjögren (MPS)
Comments by Editor Nicolas Brüggemann (NB)

Authors comments are given in black, and our responses are given in blue.

General Comments

BK - The study attempts to answer important research question of effects climate change could have on bioaerosol emissions in forests.
BK - The manuscript is well written, and scientific results and conclusions are presented in a clear, concise, and well-structured way.
MPS - I believe this manuscript addresses relevant and interesting scientific questions within the scope of BG. In general, the aim, the methods and the conclusions for the study are clearly presented. I believe the authors refer to relevant references. The paper presents a novel method for measurement of biological aerosol particles.
MPS - The abstract is clear and straight-forward. I believe the references in the introduction are relevant.
MS - The manuscript is well written, and the topic is important because low-cost optical particle counters (OPCs) are likely to have a place in aerobiological monitoring.

Response – we thank the reviewers for their positive comments.

Questions around the specificity of the low cost OPC technique towards bioaerosols, and the potential for inorganic particles present

(BK) The authors should make additional effort to assure that data collected by used optical particle counters represent bioaerosols. Notable quantity of inorganic particles should be present in the size range analysed and there is no evidence that bioaerosols dominate. A notable amount of sand particles or plant debris can be suspended in the atmosphere, especially in Autumn.

(BK) The hypotheses are focused on fungal spores, and it should be better addressed in results what is efficiency of used methodology for sampling expected diversity of fungal spores in studied environments. The authors clearly indicated that the size range chosen for the analysis (1-10 µm) is just the fraction of typical size range for fungal spores (1-30 µm).

(NB) One important issue that needs to be addressed is the selectivity or unambiguity of your measurements with respect to fungal spores. You claim that because of the location of your site (rural woodland) and time of the year (autumn), the likelihood of also having measured inorganic particles is very low. However, it would be better to provide evidence beyond correlation with the results from the CAMS global reanalysis you mentioned in your response. A very simple method

would be to analyze the loss on ignition of a particle sample. If the loss is (almost) 100%, it is (almost) all biogenic material. If there is a significant amount of inorganic particles, the LOI will be well below 100%. If you have sample material left over, it would be ideal if you could provide analytical evidence (alternatively also, e.g., scanning electron microscopic image analysis).

(MS) There are several inherent problems with the use of PM data as a proxy for bioaerosols, as there will be both organic and inorganic particles in the air, even in rural settings. Although the levels of dust in the air during autumn in the UK will not be as noticeable as other parts of the world, such as in Continental or Mediterranean climates, this should also be accounted for. It is even more problematical to say that the particles are 'fungal spores' as there are no fungal spore data available for comparison.

(BK) and (MS) question the lack of specificity of the technique towards bioaerosols and fungal spores in particular. Optical particle counters (OPCs) do not explicitly discriminate between bioaerosols and other aerosol compositions (as stated in section 2.2 of the paper). However, we believe that due to the location of the study site (a woodland in a rural location), the field measurements dates in peak fungal activity, and the low hygroscopicity of the particles we measured, it is very likely we are detecting a predominantly biological source. In broad terms, through this paper, we aim to show that the benefits and potential of the low-cost optical particle counters (OPCs) to detect bioaerosols. We also believe that we have shown, with caution not to over-interpret the data, OPC data can yield meaningful data addressing significant science questions. We note that this is the first study on bioaerosols (or aerosols) in any of the current or previous forest FACE experiments. The low cost, low power, and small spatial footprints of OPCs outweigh some of the negatives (such as the limited size range) because they enable more studies, with a greater number of measurement points, to be undertaken on forest bioaerosols.

(BK) and (MS) suggest there will be a notable amount of sand particles or plant debris present. We include plant debris in our definition of bioaerosols, as stated in the abstract, although it is likely that fungal spores will dominate the bioaerosol fraction due to the timing of the experimental duration. For the measurement period of the experiment, it is unlikely there are any great intrusions of sand particulates.

(NB) suggests the analysis of particulate matter (PM) samples collected within the forest. This is a good idea and one we hope to pursue in the future. Unfortunately, we do not have historic PM samples from the forest so currently we cannot perform these analyses.  We further explored to see if there were other physical samples collected that could be used as proxy data, but we could not find anything suitable.  To obtain PM samples during the fungal season, we would now need to wait to autumn 2022.  However, the data collected at this stage would be different from the data collected from this field campaign in 2018 and thus would not be directly comparable.
The UK does experience desert dust inclusions, especially in the Spring but less so in the autumn. To further explore this, we now provide new sections in the paper using the Copernicus Atmosphere Monitoring Service (CAMS) global reanalysis (EAC4) data as a measure of regional aerosol activity influencing the study site.  Section 2.5 provides the new methodology, and section 3.3 provides the new results. The Copernicus Atmosphere Monitoring Service (CAMS) reanalysis is the latest global reanalysis dataset of atmospheric composition produced by the European Centre for Medium-Range Weather Forecasts (ECMWF), consisting of three-dimensional time-consistent atmospheric composition fields, including aerosols and chemical species (Inness, Ades et al. 2019).

We choose the nearest available grid point from the EAC4 dataset (longitude = -2.50, latitude = 52.75) to the BIFoR site (longitude = -2.32, latitude = 52.81) and use the same time period as the

Ambient and eCO$_2$ measurements.  The new Figure (shown below, Figure 8 in the updated paper). The PM$_{10}$-PM$_1$ metric is calculated from the EAC4 estimates of PM$_{10}$ and PM$_1$.

[Figure]

*Figure 1. Time series data of the PM10-PM1 mass concentrations measured in the Ambient and eCO$_2$ arrays and provided by the Copernicus Atmosphere Monitoring Service (CAMS) global reanalysis (EAC4) dataset.  The BIFoR data contains only data collected when RH < 99%.*

It can be seen in the Figure that the EAC4 estimate of regional PM$_{10}$-PM$_1$ does not correlate with the PM$_{10}$-PM$_1$ measured within the forest. Hence, we believe the measured PM$_{10}$-PM$_1$ is local to woodland and not representative of the regional air mass. The Pearson correlation between the measured PM$_{10}$-PM$_1$ in the Ambient and eCO$_2$ arrays and the EAC4 estimates is -3 and 3, respectively, highlighting negligible correlation between the measured and EAC4 timeseries. To look for further possible correlations between the measured PM$_{10}$-PM$_1$ and estimated aerosol properties, correlative analysis was conducted on the measured PM$_{10}$-PM$_1$ with the aerosol optical depth (AOD) estimates from EAC4 consisting of the total AOD, the AOD attributed to dust aerosol, the AOD attributed to sea salt aerosol, and the AOD attributed to sulphate aerosol. All AODs were for the wavelength of 550 nm.  Similar to the EAC4 PM$_{10}$-PM$_1$ product result, all the correlations with the AOD products were negligible, with Pearson correlation values ranging from -16 to 23. See supplementary material for more information the correlation plots provided in Figures S1 to S11.

(BK) noted the OPCs as configured were only capable of measuring particles up to 10 µm in size, and therefore some larger fungal spores and pollen particles are not capable of being detected. We explicitly highlight this limitation on line 437 of the present manuscript. However, many common woodland spore species are smaller than 10 µm, including the species listed on lines 126-130 of the paper, as well as several species of airborne spores shown to be extremely common across the UK including Cladosporium, Ganoderma, and Aspergillus see (Sadyś et al., 2016).  We now state the following, where the 10 µm size limit restriction is mentioned "We note that many common woodland spore species are smaller than 10 µm, including the following species observed in the forest: *Lactarius quietus*, *Russula* species, *Lycoperdon perlatum*, *Mycena rosea,* other *Mycena* species, *Auricularia auricula*, *Hypholoma fasciculare*, *Xylaria hypoxylon*, and various *Stereum* species.

In addition to the observed species, many spores commonly observed to be airborne in the UK, including *Cladosporium*, *Ganoderma*, and *Aspergillus* (Sadyś et al., 2016), have diameters less than 10 μm."

In conclusion, we agree with the reviewers that it would be advantageous to have aerosol chemical composition measurements in addition to aerosol size measurements to unambiguously determine bioaerosol concentration, but practical constraints made that impossible in this field campaign. We hope that future field campaigns will allow for more extensive equipment payloads to be utilized. To make clearer to the reader we are measuring $PM_{10}$-$PM_1$, and that we believe that $PM_{10}$-$PM_1$ is most likely predominantly composed of bioaerosol aerosols, we make the following clarifications.

In the abstract we change "We installed low-cost optical particle counters (OPCs) to measure particles in the size range between 1 and 10 μm, where bioaerosols will likely dominate the particle mass concentration, for a period of two months in Autumn 2018 at the Birmingham Institute of Forest Research (BIFoR) Free Air Carbon Dioxide Enrichment (FACE) facility." to "In the paper, we propose that the $PM_{10}$-$PM_1$ metric is a good proxy for bioaerosols because of the bioaerosol representative size range, the location of the study site (a woodland in a rural location), the field measurement taking place during the season of peak fungal activity, and the low hygroscopicity of the particles measured."

In the conclusions we change "We have demonstrated that low-cost OPCs are suitable for measuring bioaerosol concentrations in forests, or other high-humidity environments." to "We demonstrate that the $PM_{10}$-$PM_1$ metric is a good proxy for bioaerosols because of the bioaerosol representative size range, the location of the study site (a woodland in a rural location), the field measurement taking place during the season of peak fungal activity, and the low hygroscopicity of the particles measured. Through comparison with the EAC4 estimate of $PM_{10}$-$PM_1$ we highlight that the woodland measurements do not follow regional air pollution trends and that the observed $PM_{10}$-$PM_1$ concentrations are likely from woodland sources." Furthermore, we now include the following statement "The use of the $PM_{10}$-$PM_1$ metric as a proxy for bioaerosols in woodland and other settings should be further evaluated through future experiments that unambiguously measure bioaerosol concentrations."

Response of low cost sensor to high relative humidity (RH)

(BK) Also, there is a concern that in high humidity conditions optical particle counter could detect small water droplets enhancing positive correlation between RH and particle counts.
Actually failure to validate that record from optical particle counter (PM 1-10) corresponds to total bioaerosols (if not total fungal spores) severely undermines the possibility to test research questions set by authors. Without such validation the authors have results for PM 1-10 and the relation to bioaerosols or fungal spores can only be discussed.

Response - You note the importance of the relationship between RH and particle counts. We have discussed this extensively in the paper in sections 2.2, 3.1, and in the first two paragraphs of the discussion, including citing the relevant literature on hygroscopic corrections of data from OPCs. One signature of bioaerosols is their low hygroscopicity, which is observed in this study. In the conclusions we note that additional work in characterising the kappa values required for bioaerosols would be useful future work.

(MPS) On row 256 you mentioned that RH was high throughout the measurement period, clarify what you mean by high?

Response - we have clarified this sentence.

Other comments

(BK) More details about setup of instruments would be needed. For how long particle samples were taken every 60s and what is the volume of air sampled? At what height the sensors were positioned?

Response - We describe the instrument set-up in Section 2.3, including the OPC specifications, experimental duration, and the height at which the instruments were installed. We now include the volume flow rate through the instrument (5.5 L/min)

(BK) If high resolution wind measurements are available, I would encourage authors to check also the effect of turbulent kinetic energy on particle concentrations since the atmospheric instability could have more pronounced effect on spore dispersion than the wind speed alone.

Response - this is a good idea but was beyond the scope of this paper. High-resolution 3D observations, from which the turbulence kinetic energy can be calculated, are available around the edge of the present site. However, a high density of equipment would be required to obtain spatially representative 3D velocity measurements in each study array. We hope that future field campaigns will allow for more extensive equipment payloads to be utilized, for example, to investigate the response of spore concentrations to gusts. Because of the difficulties inherent in interpreting 3D velocities near forest floors—i.e. those most relevant to woodland fungi— in this study, we do not rely on the high-resolution measurements from around the edge of the site.

(BK) Regarding the effects of eCO2 the authors should indicate what direct and what indirect effects are expected to increase fungal bioaerosol concentrations. It is not clear whether the CO2 increase is expected only in canopy layer or also at the ground layer where notable number of fungal spore sources could be growing. In my view eCO2 is expected to increase the vegetative mass of plants but such direct effect is not so straightforward for fungal spore sources since for many the growth and sporulation might have started only after the fumigation has ended. So if the most feasible is indirect effect through increase in amount leaf litter (as discussed) would not then be meaningful to have information about the quantity of leaf litter. This way the authors speculate twice: that the leaf litter is increased and that such increase relates to airborne fungal spore emission.

Response - Regarding the effects of eCO$_2$ on bioaerosol concentrations, we have discussed the possible direct and indirect effects in the final two paragraphs of the discussion (section 4) including changing fungi speciation, changing habitat and fungal substrates. This discussion includes prior work on the effect of CO$_2$ on fungi. This is the first study on aerosols or bioaerosols in a forest FACE experiment, and we hope the greater accessibility of low-cost sensors will enable more bioaerosol studies in FACE experiments in the future, providing further evidence on the direct and indirect effects of eCO$_2$. We believe this paper is an important first step to being able to understand the role of the changing CO$_2$ levels upon atmospheric bioaerosol concentrations.

(BK) Finally, since the dispersion model indicated threshold wind conditions under which mixing between plots is neglectable I suggest looking into differences in particle concentrations after wind speeds above that threshold is eliminated.

Response - 70% of measurements occurred during low, and low-medium wind speed conditions (less than 3 m s$^{-1}$), whereby the model demonstrated that mixing between arrays is highly unlikely. Even at these lower wind speeds, the effects of eCO$_2$ discussed elsewhere in the paper still apply-whereby high concentration events are suppressed by eCO$_2$, but lower background concentrations are the same between eCO$_2$ treatment and control.

(MPS) I believe you could consider including the use of low-cost sensors in the title since this is an important concept of the paper.

Response –We update the title to "Mass concentration measurements of autumn bioaerosol using low cost sensors in a mature temperate woodland Free Air Carbon Dioxide Enrichment (FACE) experiment: investigating the role of meteorology and carbon dioxide levels"

(MPS) When reading the introduction I was curious if the type of sensors you are using have ever been used in the same way before? I'm also wondering if it can be made clearer why you hypothesize that the fungal bioaerosol concentration should increase with increasing CO2?

Response – to the best of our knowledge low-cost optical particle counters have not been used in a forested environment or a FACE experiment before, so this study is a novel method. You also ask about the hypothesized mechanism by which fungal bioaerosol would increase. We believe we have listed the appropriate literature for this topic, however we now include a clarifying sentence explicitly stating the link between fungal sporocarp production, spore production, and airborne fungal bioaerosol concentrations: "All of these demonstrated changes in fungal phenology, sporocarp production, and sporulation suggest that bioaerosol concentrations are also likely to change under eCO$_2$. Even if these findings are fungal species-specific, they have potentially wide-ranging effects for forested habitats."

(MPS) The methods section is extensive. I believe the confirmation done with the macro-fungi survey is important but I'm not sure you explain this survey thoroughly: Would it be possible to extend on this matter? I work with aerosols and bioaerosols but I am not familiar with this type of macro-fungi survey. Or put in a reference?

Response – We now include an appropriate reference to improve clarity with respect to macrofungal surveying (Van Norman et al., 2008).

(MPS) In the instrumentation section you make assumptions about the particle density and the refractive index, how did you choose those? Do you have a reference for the choice?
Did you look into the literature how the presumed concentrations of bioaerosols that you measure compare to the concentrations measured with other more specific instruments (e.g. WIBS and UV-APS) in forests?

Response – we use the standard particle density and refractive index used by the OPC algorithm. Particle density of 1.65 g/ml and refractive index of 1.5 + i0 are used as default settings on the OPC device.  The literature is not very forthcoming with spore densities or refractive indices.  For pollen the estimate density is between 1000-2000 g/ml (Pope 2010).  We now highlight the implications of this good question in the test "The choice of particle density and refractive index has implications for the derived mass concentrations, however comparison between measurements taken within the woodland should be self-consistent".

We researched the literature for bioaerosol concentrations and believe we have referenced the relevant papers (lines 53 to 59), however it is challenging to make a direct comparison between bioaerosol concentrations measured in different forests because of the use of varying methodologies. As noted above, we hope that future studies will be able to perform direct instrumentation comparisons in the same location and measurement period in order to investigate this question further.

The median average $PM_{10}$-$PM_1$ mass concentration observed was 9.3 and 10.1 µg/m$^3$ under $eCO_2$ and ambient conditions. Assuming this mass only represents spores with a median spore diameter of 3 µm, density of 1.65 g/ml and sphericity this equates to median spore concentration of approximately 50,000 m$^{-3}$, which falls at the high end of Sesartic and Dallafior (2011) estimate of ground level spore concentrations (10,000-50,000 m$^{-3}$). Since the measurement was at peak fungal season in a woodland, this seems entirely possible.   We now add the following line in the text: "If the observed median concentrations of $PM_{10}$-$PM_1$, using the Crilley correction with a κ of 0.1, were solely composed of idealized spherical fungal spores with radius of 3 µm and density of 1.65 g/ml this would equate to spore concentration of approximately 50,000 spores m$^{-3}$, which falls at the high end of Sesartic and Dallafior (2011) estimate of ground level spore concentrations of 10,000-50,000 m$^{-3}$".

(MPS) In general I believe you do not report on the statistical methods used extensively enough. For instance, the Loess curve you fit is only mentioned in figure captions but not in the text. Why did you fit a Loess curve? What relationships did you expect? In row 309 a linear relationship is mentioned but I find no account of how the linearity of the relationship was assessed. What statistical tests were used to assess if there is a difference or not in row 339?  When listing your conclusions, could you include significance? For instance for the decrease in bioaerosol at lower temperatures? How was this decrease measured? In scatter plots you show prediction intervals/confidence intervals: can you explain them in the text?

Response - We appreciate your comments regarding ambiguity of the statistical methods used. We have clarified the methodologies used and the rationale for using it by adding an additional section into the methods section (Section 2.6) as follows "All data analysis was completed in R version 4.0.3 (R Core Team, 2020), with figures created using openair and ggplot (Carslaw and Ropkins, 2012; Wickham, 2016). Relationships between $PM_{10}$-$PM_1$ concentrations and RH, temperature, and wind speed were visualised used scatter plots and smoothed loess curves, generated in ggplot. Box plots with mean $PM_{10}$-$PM_1$ (and interquartile ranges) were generated to visualise differences in bioaerosol concentrations between $eCO_2$ and ambient arrays. Scatter plots with regression lines were generated for Figure 7.  Kruskal-Wallis tests were used to test for statistically significant differences between the $eCO_2$ and Ambient arrays in Section 3.2."

(MPS) Text in Figure 2-Figure 7 are too small. Legends are also missing and should be put in. Make sure you clearly explain the difference between colors in the plots. For Figure 2 a difference in color intensity is mentioned in the caption but I can't detect this intensity difference, can you clarify this? For panel E and F in Figure 2, is the data hourly or daily? What is the resolution for the data displayed in Figure 3A?

We have updated the figures as requested.

(MPS) At row 296-297, there is a sentence "Detecting events..", is this a conclusion you're making about the sensors? I'm not sure I follow the reasoning, can you extend on it?

Response - the use of "events" refers to periods of time with high PM recorded, presumed as high sporulation events, which is described on line 294, we clarify the two sentences to: "Peaks in bioaerosol concentration (presumed high sporulation events) are visible in red, with some events being replicated across both $eCO_2$ treatment and control (e.g. the SW quadrant event in A5 and A6), and other high PM events only occurring in a single array (e.g. the SE quadrant event in A4). By detecting high PM events in a single array at a distinct time shows that the OPCs can detect differences between the BIFoR FACE arrays.".

(MPS) When listing your conclusions, could you include significance? For instance for the decrease in bioaerosol at lower temperatures? How was this decrease measured btw?

Response – we have added the statistical significance values into the conclusions. We have described the relationship between temperature and PM values is described in lines 308 – 311 and Figure 6, and we have summarised these results in the conclusions.

(MPS) r. 34: Missing/redundant parenthesis bracket?

Response – bracketing is correct.

(MPS) r. 93 (first row of method) add , UK. after "Staffordshire".
Figure 1: can you possibly mark the array pairs in some way? (to make it obvious that they are pairs)

Response - done

(MPS) r. 145 Perhaps you would like to include a reference for the OPC Mie scattering since you are mentioning this

Response - done

(MPS) r. 177 reference missing

Response - done

(MPS) r. 177 maybe you can refer to Table 1 here so it's clear that the OPC:s were moved around among the different arrays.

Response - done

(MPS) Table 1: To make the table easier to read, can possibly leave out the year and just have month and day for the date range?

Response - done

(MPS) Figure 4: Reference is missing here. Also check that the caption is correctly written.

Response - done

(MPS) Figure 5: Can these figures be adjusted to fit into one and the same frame. As they are now presented they take up a lot of space.

Response - done

(MPS) r. 343 reference missing.

Response - done

**Reference**

Inness, A., et al. (2019). "The CAMS reanalysis of atmospheric composition." Atmospheric Chemistry and Physics **19**(6): 3515-3556.

---

## Author Response (AR2)

**Reviewer 1:**

You have errors with references that you need to fix.

All reference errors have been corrected.

Are you clearly showing how you assessed linear relationships? For instance, when you claim that there is a linear positive relationship between temp and bioaerosol concentration (row 355)? How did you assess this relationship? Is it significant

The text in this section has been amended to reflect that the linearity of the relationship between temp and bioaerosols was not assessed.

What supports that your measurements are dominated by bioaerosols? You claim that what the sensors are predominantly measuring is bioaerosols, but what is the rest? I'm not sure you are clearly discussing this. Are you clearly presenting what other types of particles that the sensors are measuring?

We believe we have discussed this in detail in several sections in the paper. For example at line 161 we discuss how our methodology selects for predominantly bioaerosols: *"The sensors do not explicitly discriminate between particle types, so in order to discriminate between fungi and other smaller particles (bacteria and anthropogenic aerosols), we excluded data from particles smaller than 1 μm in diameter, measuring from 1 μm up to the maximum 10 μm measuring capacity of the OPCs. This size discrimination, in conjunction with the experimental location and seasonal timing of the experiment make it highly likely the majority of bioaerosols being captured are predominantly of fungal origin."*

Additionally, in the discussion (line 441), we discuss the relationship observed between particle swelling and relative humidity, and how the threshold for swelling demonstrates we are most likely measuring a biological source: *"We found that the RH % threshold for significant particle swelling was 90–95%, which is a much higher RH value than would be expected for anthropogenic aerosols, which typically contain more hygroscopic components including salts, and therefore provides evidence that the measured $PM_{10}$–$PM_1$ fraction represents a predominantly biological source. In their study investigating effects of RH on fungal spore swelling, Reponen et al. (1996) demonstrated a similar effect whereby a significant swelling of fungal spores was seen, but only at very high humidities (greater than 90%). We believe this threshold for particle swelling further demonstrates that we are recording a biological source. This hygroscopic evidence is in addition to the ecological and phenological evidence for spores being the dominant source within the $PM_{10}$-$PM_1$ size fraction during the measurement period."*

In the caption of Figure 2 you say that Kappa (with symbol) is changed, but the text in the titles this variable is indicated with the letter K. You should be consistent here: is it kappa or the letter k?

We have checked the text to ensure that kappa is represented using the "κ" symbol. The kappa symbols in Figure 2 have also been replaced to match those in the text.

Can you check that the caption of figure 4 is correctly written?

We have amended the caption of figure 4 to improve clarity.

I think figure 5 is still taking up too much space. Can you merge the figures in there? Consider re-sizing the figures.

The spacing on the figure has been improved to decrease the vertical size of the figure.

You say you used Kruskal-Wallis tests. I'm assuming this is because your data are not normally distributed, but you are not showing this (or stating why you use Kruskal-Wallis). Why?

We have added additional explanation at line 283 explicitly state that the data was non-normal.

A period is missing between "self-consistent" and "It" at row 157.

This has been corrected.

**Reviewer 2:**

The authors examined bioaerosol using a low-cost sensor in a FACE experiment filed. The OPC-measured particles were assumed as bioaerosols and the relations with eCO2 experiment and meteorological parameters were investigated. They concluded that there is no effect of eCO2 on bioaerosol concentrations. The writing is generally well structured and necessary figures provided. The authors well revised the manuscript taking the comments from the previous round of review. I suggest a minor revision, while the following points need to be further clarified.

Major point:

The OPC-measured particles could reflect dust from the soil near the OPC location as well. As there are no correlations of PM10-PM1 concentration with CAMS AOD, the observed value could represent a very small niche, e.g., centimeter to meter levels, in the forest. Consider the observation height of 1.5m in the site, suspension of soil dust could be a large source of the OPC-measured particles, especially under the wind speed of 2.59 m/s below canopy (L331-332). In such case, instead of eCO2, the degree of land cover near the sensors and the corresponding wind resistance could be more relevant. A separation, justification, or declaration of the caveat regarding the possible mis-catching of dust should be provided.

Minor points:

1. L266-270, the information of the grid resolution of CAMS is necessary.

We have added the grid reference on line 269. "The spatial resolution of the reanalysis data is 80 km"

2. The OPC measurement height was 1.5m at L206-207, but 2m in Fig4 caption. Which one is correct?

For the side-by-side intercomparison period detailed in line 206, the OPCs were installed at 1.5m height, however for the main experiment they were installed at 2m in order to best match the below-canopy wind speed and direction sensors (also installed at 2m height). We have added an extra clarification of the OPC height for the main experiment on line 186.

3. L421-423, what do -3 and 3 stand for? Pearson correlation coefficient should be less than 1. Same for L427-428.

Good spot! The openair package multiplies the Pearson coefficient by 100 to make the correlation plot easier to read. Hence –3 and 3 equal –0.03 and 0.03, respectively. We have updated the numbers in the text and put a note on the correlation plot in the supplementary material.

4. Figure legends are needed in Fig 6b and Fig 8.

We have added the appropriate figure legends for Figure 6b and Figure 8.

5. What type of regressions are used, and the lines represent in Figs. 2A-C, 3B-C, 7C, 7D, 7F, and Fig. S2-S11? What do the shades nearby each of the fitted lines mean?

We have included details of the regressions used in section 2.6 on Line 278: "Relationships between $PM_{10}$-$PM_1$ concentrations and RH, temperature, and wind speed (**Error! Reference source not found.**, **Error! Reference source not found.**, and **Error! Reference source not found.**) were visualised used scatter plots and smoothed loess curves, generated in ggplot. Box plots with mean $PM_{10}$-$PM_1$ (and interquartile ranges) were generated to visualise differences in bioaerosol concentrations between $eCO_2$ and ambient arrays. Scatter plots with regression lines were generated for Figure 7 and for the supplementary figures."

---

## Author Response (AR3)

Dear Editor,

We have updated the file.

Many thanks for your help with this paper.

Best Wishes,

Francis